# Bacterial-induced or passively administered interferon gamma conditions the lung for early control of SARS-CoV-2

Kerry L. Hilligan [1,2] ✉, Sivaranjani Namasivayam [1], Chad S. Clancy [3], Paul J. Baker[4], Samuel I. Old [2], Victoria Peluf [1,5], Eduardo P. Amaral [1], Sandra D. Oland [1], Danielle O'Mard[1], Julie Laux [6], Melanie Cohen [6], Nicole L. Garza[7], Bernard A. P. Lafont [7], Reed F. Johnson[7], Carl G. Feng [8,9], Dragana Jankovic [1,5], Olivier Lamiable [2], Katrin D. Mayer-Barber [4] & Alan Sher [1] ✉

Type-1 and type-3 interferons (IFNs) are important for control of viral replication; however, less is known about the role of Type-2 IFN (IFNγ) in anti-viral immunity. We previously observed that lung infection with *Mycobacterium bovis* BCG achieved though intravenous (*iv*) administration provides strong protection against SARS-CoV-2 in mice yet drives low levels of type-1 IFNs but robust IFNγ. Here we examine the role of ongoing IFNγ responses to pre-established bacterial infection on SARS-CoV-2 disease outcomes in two murine models. We report that IFNγ is required for *iv* BCG induced reduction in pulmonary viral loads, an outcome dependent on IFNγ receptor expression by non-hematopoietic cells. Importantly, we show that BCG infection prompts pulmonary epithelial cells to upregulate IFN-stimulated genes with reported anti-viral activity in an IFNγ-dependent manner, suggesting a possible mechanism for the observed protection. Finally, we confirm the anti-viral properties of IFNγ by demonstrating that the recombinant cytokine itself provides strong protection against SARS-CoV-2 challenge when administered intranasally. Together, our data show that a pre-established IFNγ response within the lung is protective against SARS-CoV-2 infection, suggesting that concurrent or recent infections that drive IFNγ may limit the pathogenesis of SARS-CoV-2 and supporting possible prophylactic uses of IFNγ in COVID-19 management.

COVID-19 is a pulmonary disease caused by SARS-CoV-2 which infects lung epithelial cells via the membrane protein angiotensin-converting enzyme 2 (ACE2)[1,2]. ACE2 is highly expressed by pneumocytes and particular subsets of ciliated bronchial cells[3,4], thus making these cell types the primary target for SARS-CoV-2 infection[5]. Interferons (IFNs) play a central role in anti-viral immunity, through the induction of host defense elements that constrain viral invasion, replication, and release in target epithelial cells[6,7], as well as by facilitating immune cell activation and recruitment[8]. Type-1 IFNs (IFN-I, including IFNα and IFNβ) and type-3 IFNs (IFNλ) are typically associated with responses against viruses and have been identified as mediators of host defense against SARS-CoV-2[9–14]. However, the role of type-2 IFN (IFNγ) during viral infection, and in particular, SARS-CoV-2 infection, is less clear.

IFNγ is a key mediator of immunity to intracellular microbes and is strongly induced upon bacterial infection. Natural killer (NK) and innate lymphoid cells contribute innate sources of IFNγ, whereas

CD4+ Th1 cells and CD8+ T cells are major producers of this cytokine later in infection. An important function of IFNγ is to arm myeloid cells with microbiocidal properties such as induction of nitric oxide synthase (NOS)−2, which can also inhibit some viruses[15]. In addition, IFNγ broadly induces a suite of interferon-stimulated genes (ISGs), many of which are also induced by type-1 and type-3 IFNs and have been reported to also possess anti-viral activity[16]. While IFNγ is not generally required for host resistance to a pulmonary viral infection[17–21], recombinant (r)IFNγ treatment has been shown to confer protection against certain viral pathogens in animal model studies[22–26]. In COVID-19 patients, including immunocompromised individuals, rIFNγ treatment was shown to be well tolerated[27,28] and in one study was reported to reduce the time to hospital discharge[29].

Pre-clinical studies and clinical trials assessing the efficacy of therapeutics that act through IFNs have shown that the timing of treatment relative to viral exposure is crucial[25,30–32]. Successful regimens usually involve treatment before or just after viral exposure limiting their potential therapeutic use. However, these observations do raise some interesting questions concerning host susceptibility to SARS-CoV-2 infection in the context of an ongoing IFN response to concurrent or recent infection. Indeed, we and others have previously observed that lung infection with *Mycobacterium bovis* BCG achieved through intravenous (*iv*) administration provides protection against SARS-CoV-2 in mouse and hamster models[33–36]. Likewise, aerosol *Mycobacterium tuberculosis* infection has been shown to be associated with lower SARS-CoV-2 viral burdens and improved survival in mice[37–39].

Here we explored the mechanisms by which concurrent bacterial infection protects against SARS-CoV-2 and show that IFNγ is an essential mediator of BCG conferred protection in vivo. IFNγ was found to act on epithelial cells to limit SARS-CoV-2 infection and/or replication, possibly through the induction of anti-viral proteins. Intranasal administration of the recombinant cytokine prior to the viral challenge also elicited strong protection against SARS-CoV-2, recapitulating our observations in bacterial co-infection models. Together these observations support a role for pre-existing IFNγ responses in mediating early control of SARS-CoV-2.

## Results

### *iv* BCG alters the pulmonary cellular landscape and promotes a strong IFNγ signature

We and others have observed that intravenous (*iv*), but not subcutaneous (*sc*), administration of BCG protects against SARS-CoV-2 in mice and hamsters[33–36], thus providing a platform to dissect mechanisms of host resistance to SARS-CoV-2. To gain a deeper understanding of *iv* BCG-mediated protection against SARS-CoV-2, we set up a new series of experiments where BCG or PBS was administered *iv* to wildtype (WT) B6 mice 40–45 days before SARS-CoV-2 infection, at which time there are -$10^5$ BCG CFU present in the lung[33]. In these experiments, we utilized a beta variant (B.1.351) of the virus carrying an N501Y mutation that can transiently infect WT mice through binding of the endogenous murine ACE2 receptor[40–44] (Fig. 1A). As we had previously observed with an alpha variant of SARS-CoV-2[33], *iv* BCG significantly reduced lung viral burden at 3 days after challenge with a beta variant (Fig. 1B). To identify correlates of protection, we performed single-cell RNA sequencing (scRNAseq) on lung cells isolated from the same set of animals at the same timepoint. Seurat clustering revealed 28 distinct cell types encompassing epithelial, endothelial, stromal, myeloid, and lymphoid lineages (Fig. 1C, Fig. S1A, and Supplementary Data 1). When comparing the abundance of different clusters between control (PBS) and *iv* BCG animals, we found CD4+ T cells and CD8+ T cells to be enriched after *iv* BCG and fibroblast subsets to be comparatively higher in controls (Fig. 1D). This is in line with flow cytometry data showing significantly higher numbers of T lymphocytes present in the lung tissue of *iv* BCG mice

compared to controls, which is also apparent prior to SARS-CoV-2 challenge (Fig. S1B, C).

We next performed differential expression analysis of scRNAseq data which showed that T cell and myeloid cell clusters, in particular macrophage populations mac(1), mac(2), and dendritic cells (DC), had the highest number of differentially expressed genes (DEGs) between PBS and BCG-treated mice (Fig. 1E and Supplementary Data 2). Sub-clustering of the myeloid compartment identified 16 clusters encompassing macrophages, monocytes, dendritic cells, neutrophils, and mast cells, many of which showed prominent condition-specific clustering (Fig. 1F, Fig. S1D and Supplementary Data 3). Notably, resident alveolar macrophages were found exclusively in samples from control animals, whereas *iv* BCG administration was associated with monocyte-derived macrophages (Fig. 1F). In addition to genes related to cell ontogeny (Fig. S1E), DEGs between these two macrophage populations were related to inflammatory responses, anti-viral signatures and interferon signaling, with the alveolar macrophages from control animals expressing an IFN-I signature (*Isg15*, *Oas3*, *Ifitm3*) consistent with the high viral titers recovered from the lungs. In contrast, the monocyte-derived macrophages from BCG-treated mice expressed robust glycolytic, antigen presentation, and IFNγ signatures including high levels of *Nos2*, *Cxcl9*, and numerous MHCII-related genes (Fig. 1G, Fig. S1E, and Supplementary Data 4). This observation is in line with cytokine multiplex and ELISA data from SARS-CoV-2 naïve animals showing that IFNα and IFNλ3 levels did not differ between PBS and BCG groups, whereas IFNβ and IFNγ were significantly higher in samples from BCG-treated animals compared to controls, with the most striking increase seen in the IFNγ levels (Fig. 1H).

To evaluate the sources of IFNγ after *iv* BCG, we performed intracellular cytokine staining on lung single-cell suspensions 28 days following BCG inoculation. This confirmed that IFNγ+ cells were enriched in the lungs of mice following *iv* BCG in the absence of SARS-CoV-2 infection and that CD4+ and CD8+ T cells, as well as NK cells, were the predominant sources of the cytokine (Fig. 1I). While these data are at 28 days post BCG, they are consistent with flow cytometric analysis showing increased CD4+ T cell expression of the Th1 master transcription factor Tbet 42 days after *iv* BCG (the time when animals are challenged with SARS-CoV-2) (Fig. S1F). Next, we mapped *Ifng* transcripts against the UMAP projection of the single-cell RNAseq clustering. *Ifng* was predominantly expressed by CD4+ and CD8+ T lymphocytes as well as NK cells in *iv* BCG mice challenged with SARS-CoV-2. Only low-level *Ifng* expression was observed in mice infected with SARS-CoV-2 alone and was largely restricted to CD8+ T cells and NK cells (Fig. S1G).

Together these data show that *iv* BCG induces a T and NK cell driven IFNγ response in the lung, which is apparent before and after SARS-CoV-2 challenge. Importantly, IFNγ levels and the frequency of Tbet+ Th1 cells were significantly higher after *iv* administration compared to *sc* BCG inoculation, which we have previously shown is unable to protect animals against SARS-CoV-2 (Fig. 1I and Fig. S1F)[33]. This route-dependent induction of Th1 cells producing IFNγ is likely due the fact that only *iv* administration establishes a substantial bacterial infection of the lung[33]. We therefore hypothesized that a bacteria prompted IFNγ response within the pulmonary micro-environment prior to SARS-CoV-2 exposure is involved in the protection conferred by *iv* BCG.

### IFNγ is required for *iv* BCG conferred protection against SARS-CoV-2

We next evaluated whether IFNγ receptor signaling and T cells, the major source of IFNγ after *iv* BCG, are required for protection against SARS-CoV-2 in response to *iv* BCG. To this end, BCG was administered *iv* to WT B6, *Ifngr1*$^{-/-}$ or *Tcra*$^{-/-}$ mice 40–45 days before intranasal SARS-CoV-2 challenge. At three days post SARS-CoV-2 challenge, viral

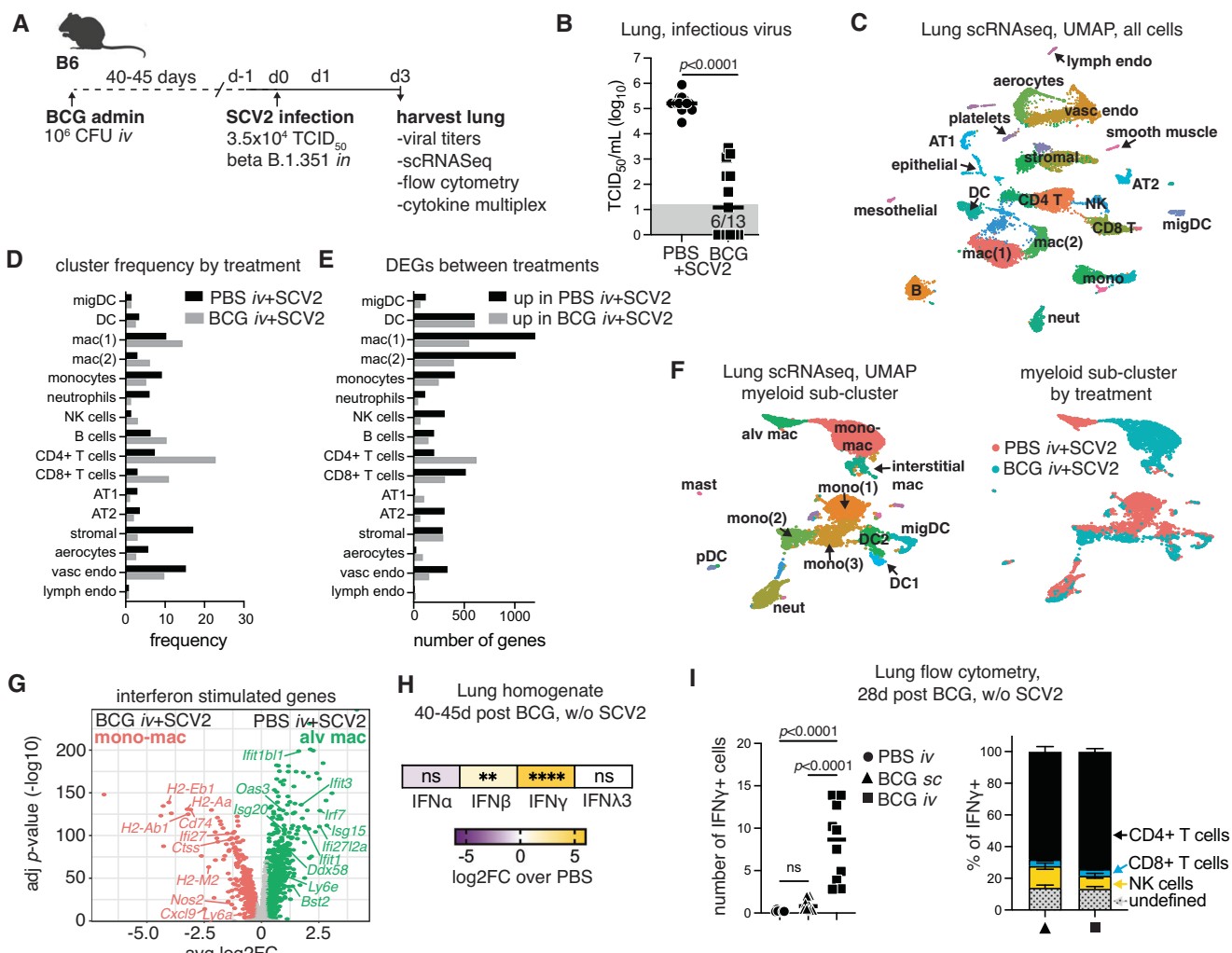

**Fig. 1 | *iv* BCG skews the pulmonary cytokine landscape towards IFNγ production.** B6 mice were inoculated with BCG or PBS *iv* 40–45 days prior to intranasal challenge with SARS-CoV-2 (SCV2) B.1.351. Lungs were harvested 3 days after viral challenge. **A** Schematic of the experimental protocol. **B** Viral titers in lung homogenate as measured by TCID$_{50}$ assay (PBS *n* = 14, BCG *n* = 13; pooled from two independent experiments; two-tailed Mann–Whitney test). Gray box shows values below limit of detection. **C** UMAP representation of scRNAseq data of the whole lung isolated from BCG and PBS animals challenged with SARS-CoV-2. Clustering resolution: 0.6. Cluster-specific annotation is shown in Fig. S1A and gene sets in Supplementary Data 1. **D** Frequency of cells in each cluster separated by experimental condition. **E** Number of differentially expressed genes (log2FC > 0.25, *p* < 0.05, Wilcoxon Rank-Sum test with Bonferroni correction) between PBS and BCG groups for each cluster. Gene lists with their associated FC and *p* values are in Supplementary Data 2. **F** UMAP representation of sub-clustered myeloid cells for PBS or BCG-treated animals (left) and UMAP colored by treatment group (right).

Clustering resolution: 0.4. Cluster-specific annotation is shown in Fig. S1D and gene sets in Supplementary Data 3. **G** Volcano plot shows DEGs with annotated ISGs between resident AM and monocyte-derived macrophage clusters (log2FC > 0.25, *p* < 0.05, Wilcoxon Rank Sum test with Bonferroni correction). The full gene list with their associated FC and *p* values are in Supplementary Data 4. **H** Fold change in interferon protein levels in lung homogenate between PBS and BCG-treated mice without SARS-CoV-2 challenge (PBS *n* = 13, BCG *n* = 12; pooled from three independent experiments; one-way ANOVA with Tukey post-test). Not significant (ns) *p* > 0.05; **p* = 0.0018; ****p* < 0.0001. **I** Flow cytometry data showing the number of total IFNγ+ cells (left panel) and the cellular composition of IFNγ+ cells (right panel, mean ± SEM) isolated from the lungs of mice 28 days after BCG was administered by subcutaneous (*sc*, triangle) or *iv* injection (square) (PBS *n* = 8, BCG sc *n* = 10, BCG iv *n* = 10; pooled from two independent experiments; one-way ANOVA with Tukey post-test). Not significant (ns) *p* > 0.05. Source data are provided as a Source Data file.

---

loads were assessed in the lung by tissue culture infectious dose-50 (TCID$_{50}$) assay (Fig. 2A). Strikingly, in the absence of the IFNγ receptor, prior *iv* BCG infection failed to reduce viral loads as seen in WT B6 mice. Similarly, BCG protection was diminished in T cell-deficient animals, although some reduction in viral load was still noted, potentially due to residual IFNγ produced by NK cells in these animals (Fig. 2B)[45]. Mice with deficiencies in IFNγ signaling and/or T cells are highly susceptible to BCG[45,46] and consistent with this, the colony-forming units (CFU) recovered from the lungs of *Ifngr1*[−/−] or *Tcra*[−/−] animals were significantly higher than WT B6 controls (Fig. S2). To control for this difference in CFU burden, we adopted an alternative approach where WT B6 mice were inoculated *iv* with BCG,

and IFNγ was subsequently neutralized through intraperitoneal administration of an anti-IFNγ antibody starting just 1 day prior to SARS-CoV-2 challenge (Fig. 2C). Under these experimental conditions, no increase in BCG CFU was observed in the lungs after IFNγ neutralization (Fig. S2), yet SARS-CoV-2 viral loads were strikingly higher than in isotype control animals inoculated *iv* with BCG (Fig. 2D). As some residual protection was observed in *iv* BCG mice after IFNγ neutralization, we also blocked the IFN-I receptor (IFNAR) immediately prior to SARS-CoV-2 challenge to determine whether the low levels of IFNβ detected after *iv* BCG could also be contributing to the anti-SARS-CoV-2 response. This treatment failed to significantly further reduce the protection achieved by

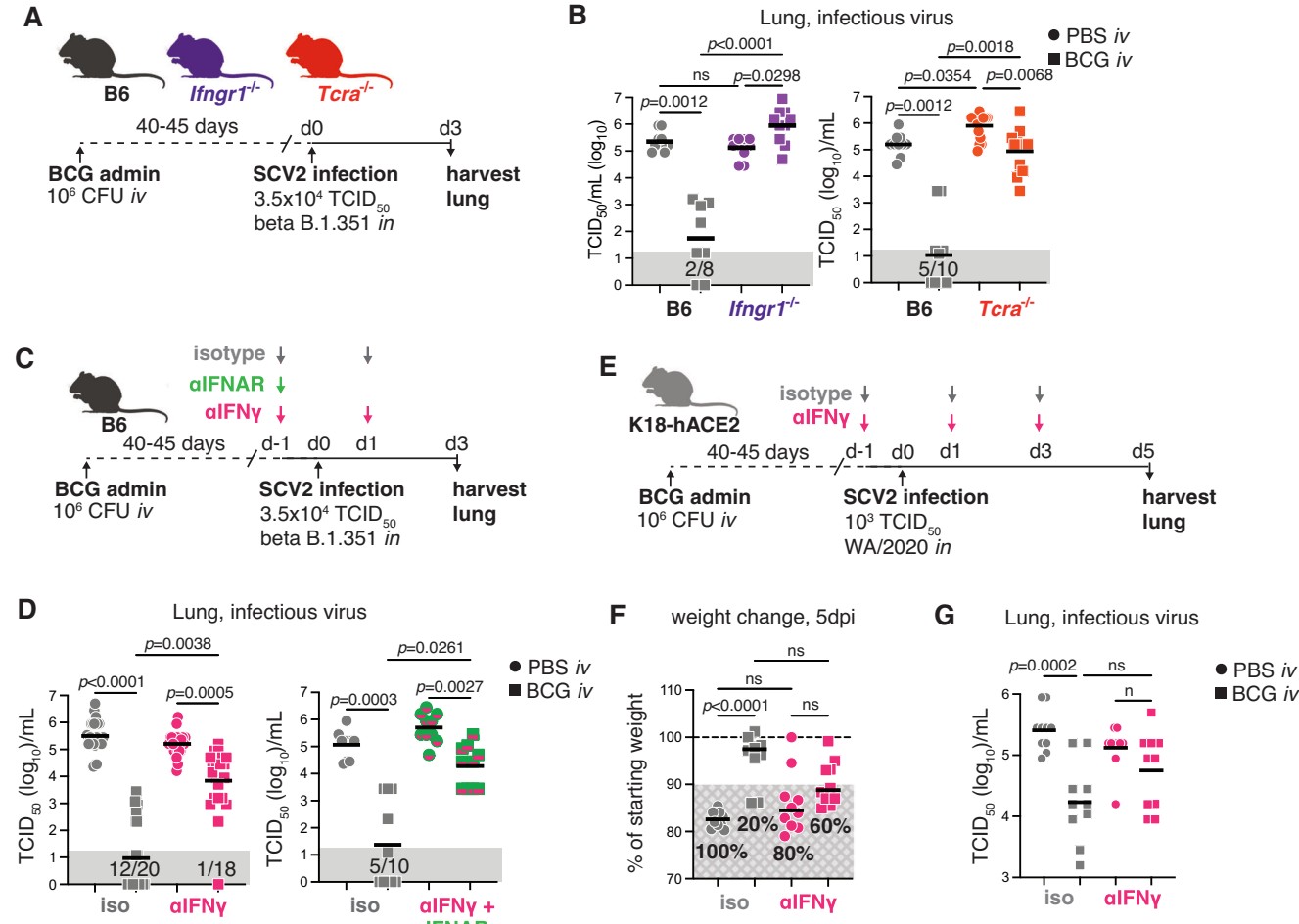

**Fig. 2 | IFNγ is required for *iv* BCG-induced protection against early SARS-CoV-2 infection.** Mice of the indicated genotypes were inoculated with BCG or PBS *iv* 40–45 days prior to intranasal challenge with SARS-CoV-2 (SCV2). Lungs were harvested 3–5 days after viral challenge. **A** Schematic of experimental protocol. **B** Viral titers in lung homogenate from B6, *Ifngr1⁻/⁻* or *Tcra⁻/⁻* mice as measured by TCID$_{50}$ assay 3 days after viral challenge (Left panel: B6 PBS $n = 10$, B6 BCG $n = 8$, *Ifngr1⁻/⁻* PBS $n = 10$. *Ifngr1⁻/⁻* $n = 10$. Right panel: B6 $n = 10$/group, *Tcra⁻/⁻* PBS $n = 10$, *Tcra⁻/⁻* BCG $n = 9$. Both panels: pooled from two independent experiments; Kruskal-Wallis with Dunn's post-test). **C** Schematic of experimental protocol with anti-IFNγ and anti-IFNAR treatment. **D** Viral titers in lung homogenate as measured by TCID$_{50}$ assay 3 days after viral challenge (Left panel: PBS isotype $n = 24$, BCG isotype $n = 24$,

PBS αIFNγ $n = 23$, BCG αIFNγ $n = 22$; pooled from 5 independent experiments. Right panel: $n = 10$/group; pooled from two independent experiments. Both panels: Kruskal–Wallis with Dunn's post test). **E** Schematic of the experimental protocol with anti-IFNγ treatment in the K18-hACE2 mouse model. **F** Percentage of starting weight at 5 dpi. Hashed gray box shows percentage of animals with greater than 10% body weight loss. **G** Viral titers in lung homogenate as measured by TCID$_{50}$ assay 5 days after SARS-CoV-2 WA/2020 challenge ($n = 10$/group; pooled from two independent experiments; Kruskal–Wallis with Dunn's post-test). Not significant (ns) $p > 0.05$. Gray boxes denote values below the limit of detection. Source data are provided as a Source Data file.

administration of the anti-IFNγ antibody alone (Fig. 2D). Together with the data showing complete loss of protection in *Ifngr1⁻/⁻* mice, this result implicated IFNγ as the dominant mediator of *iv* BCG conferred protection against SARS-CoV-2.

We next tested whether IFNγ is required for *iv* BCG-induced protection in transgenic K18-hACE2 mice which provide a model of SARS-CoV-2-induced pathology and severe disease (Fig. 2E)[47]. *iv* BCG protected K18-hACE2 mice against weight loss and resulted in lower viral loads, but no significant protection was observed in animals treated with anti-IFNγ (Fig. 2F, G). While a direct comparison between isotype and anti-IFNγ treated mice inoculated with BCG prior to viral challenge did not reach statistical significance, animals treated with anti-IFNγ showed a clear trend towards more severe disease and higher viral loads (Fig. 2F, G). Together, these data from two different in vivo models demonstrate a key role for *iv* BCG-induced IFNγ in mediating protection against SARS-CoV-2 infection. Furthermore, the observation that a short window of IFNγ neutralization is sufficient to significantly abrogate protection suggested that the cytokine mediates its effects at the time of initial SARS-CoV-2 exposure.

### *iv* BCG-induced IFNγ promotes expression of viral restriction factors but is not required at the time of SARS-CoV-2 challenge to limit virus-induced inflammation

In addition to reducing viral loads, *iv* BCG limits SARS-CoV-2-driven lethality and hyperinflammation[33]. To examine whether IFNγ is also involved in *iv* BCG-induced suppression of inflammation following SARS-CoV-2, we performed a cytokine multiplex assay, flow cytometry and scRNAseq on lung samples from B6 mice 3 days after challenge with a SARS-CoV-2 beta variant in the presence or absence of IFNγ neutralization (Fig. 3A). While WT B6 mice did not develop lethal disease, they did respond with a characteristic SARS-CoV-2 pro-inflammatory response consisting of heightened IFN-I, IFNλ, IL-6, GM-CSF, and CCL2 levels (Fig. 3B, C and Fig. S3A). These responses were absent or significantly lower in *iv* BCG inoculated animals irrespective of αIFNγ treatment (Fig. 3B and Fig. S3). Similar observations were apparent at the transcript level for *Il6*, *Csf2* (encoding GM-CSF), *Ccl2*, and *Il18* across distinct cell lineages (Fig. 3C), suggesting that dampening of SARS-CoV-2-driven hyperinflammation by *iv* BCG occurs independently of IFNγ when this cytokine is neutralized at the time of

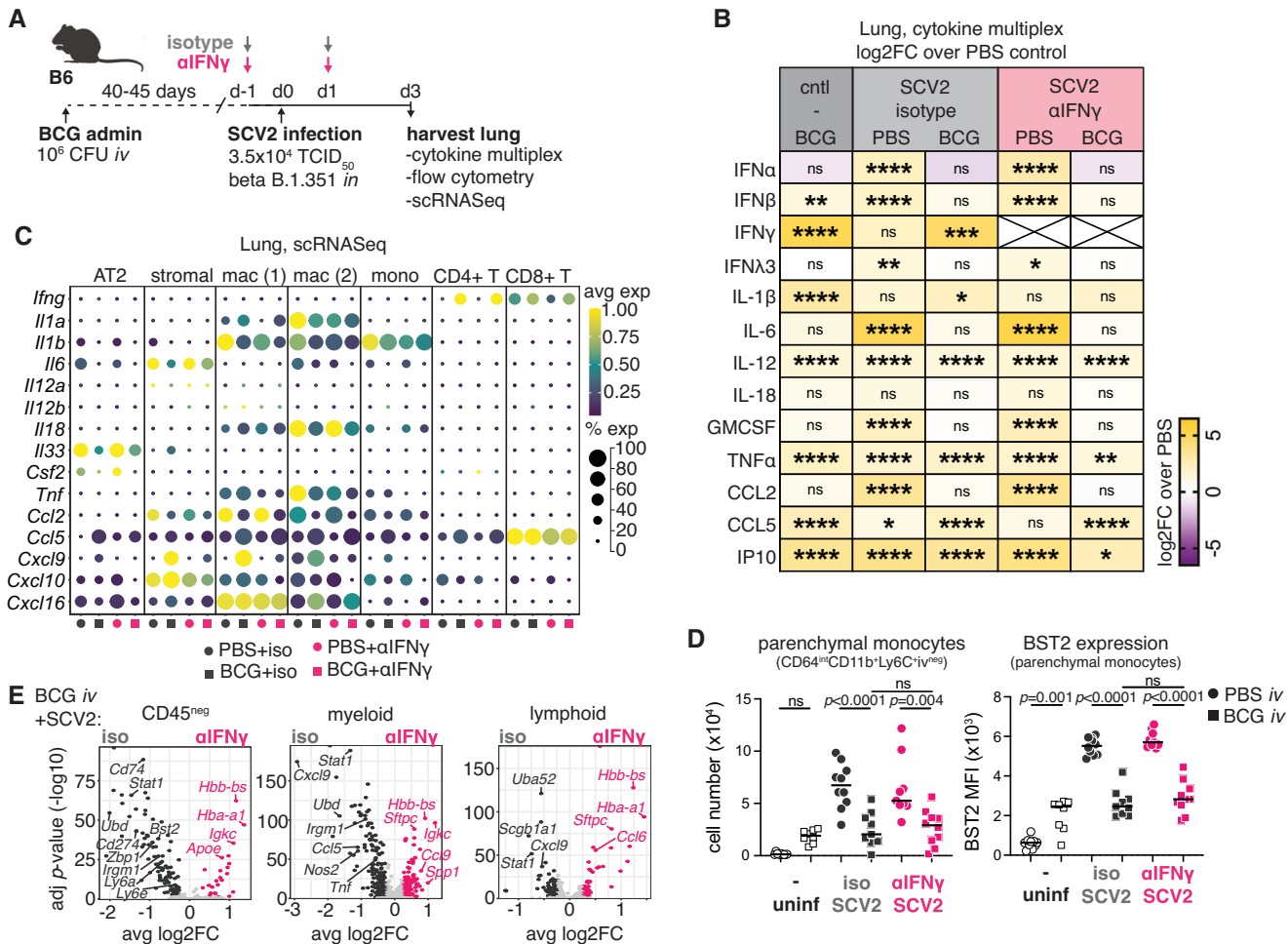

**Fig. 3 | *iv* BCG-induced IFNγ present at the time of SARS-CoV-2 challenge limits viral replication but not SARS-CoV-2-induced inflammatory responses.** B6 mice were inoculated with BCG or PBS *iv* 40–45 days prior to intranasal challenge with SARS-CoV-2 (SCV2) B.1.351. Infected animals received an IFNγ neutralizing antibody or isotype control 1 day prior to and 1 day following SARS-CoV-2 instillation. Lungs were harvested 3 days after the viral challenge. **A** Schematic of experimental protocol. **B** Heat map display of log2 fold change (FC) of cytokine levels in lung homogenate relative to PBS uninfected controls. Cytokines were measured by multiplex assay and normalized to total protein content (BCG *n* = 12, PBS + SCV2+iso *n* = 15, BCG + SCV2+iso *n* = 14, PBS + SCV2 + αIFNγ *n* = 13, BCG + SCV2 + αIFNγ *n* = 13; pooled from three independent experiments; One-Way ANOVA with Tukey post-test). Not significant (ns) *p* > 0.05; *\*p* < 0.05; *\*\*, p* < 0.01; *\*\*\*p* < 0.001; *\*\*\*\*p* < 0.0001. Raw data and *p* values are in Fig. S3. **C** Dot plot shows the relative

expression and frequency of the indicated genes across selected Seurat clusters defined in the UMAP in Fig. 1C. **D** The number of parenchymal monocytes (live/CD45⁺/Ly6G⁻/CD64ⁱⁿᵗ/CD88ⁱⁿᵗ/CD26⁻/CD11b⁺/Ly6C⁺/iv⁻) and their expression of BST2 (MFI) as determined by flow cytometry (PBS *n* = 10, BCG *n* = 8, PBS + SCV2+iso *n* = 10, BCG + SCV2+iso *n* = 9, PBS + SCV2 + αIFNγ *n* = 9, BCG + SCV2 + αIFNγ *n* = 10; pooled from two independent experiments; one-way ANOVA with Tukey post-test). Not significant (ns) *p* > 0.05. **E** Volcano plots show DEGs between isotype and anti-IFNγ treated mice inoculated *iv* with BCG across CD45ⁿᵉᵍ, myeloid, and lymphoid lineages that were manually annotated from the Seurat clustering showing in Fig. 1C. DEGs are shown in dark gray or pink (log2FC > 0.25 and *p* < 0.05, Wilcoxon Rank Sum test with Bonferroni correction). Light gray points denote genes that did not reach statistical significance. Gene lists and their associated FC and *p* values are in Supplementary Data 5–7. Source data are provided as a Source Data file.

viral challenge. This hypothesis was further supported by the reduction in virus-driven tissue infiltrating inflammatory monocytes and their expression of the IFN-inducible marker bone marrow stromal cell antigen-2 (BST2, aka Tetherin/CD317) in *iv* BCG inoculated mice with or without IFNγ neutralization (Fig. 3D).

Interestingly, despite the 4-log fold increase in viral load observed after IFNγ blockade in BCG inoculated mice (Fig. 2D), there were very few differences in the pulmonary cytokine milieu, other than the known IFNγ regulated cytokines, CXCL9, IP-10 (CXCL10) and TNFα (Fig. 3B, C). *Tnf* transcript in monocytes was reduced upon IFNγ neutralization in BCG inoculated animals as were *Cxcl9* and *Cxcl10* levels in macrophage and stromal populations (Fig. 3C). Differential expression analysis of pooled CD45-negative, myeloid or lymphoid cells revealed enrichment in ISGs (*Stat1, Ly6a, Irf1, Irgm1,* and *Nos2*) in isotype treated BCG animals, confirming effective neutralization of IFNγ in mice treated with anti-IFNγ (Fig. 3E and Supplementary Data 5–7). Notably,

genes with known anti-SARS-CoV-2 activity (*Bst2, Udb, Zbp1,* and *Ly6e*)[48–50] were among the transcripts downregulated by IFNγ neutralization within the CD45-negative pool (Fig. 3E and Supplementary Data 5–7). This downregulation of ISGs was not apparent in PBS controls treated with anti-IFNγ (Fig. S4A and Supplementary Data 8–10), consistent with the relatively low levels of IFNγ present during the early phase of SARS-CoV-2 infection (Fig. 3B). Single-cell gene set enrichment analysis (scGSEA) using a manually curated list of genes with experimentally validated SARS-CoV-2 restriction activity (Supplementary Data 11)[48,49] supported these observations. Despite the increase in viral load, IFNγ neutralization significantly reduced the enrichment of the "anti-SARS-CoV-2" gene set across numerous cell clusters, including pulmonary epithelial cells, in mice inoculated with BCG prior to viral challenge (Fig. S4B, C). Overall, these results suggest that *iv* BCG-induced IFNγ present at the time of viral challenge acts by directly controlling viral load rather than SARS-CoV-2-driven inflammation.

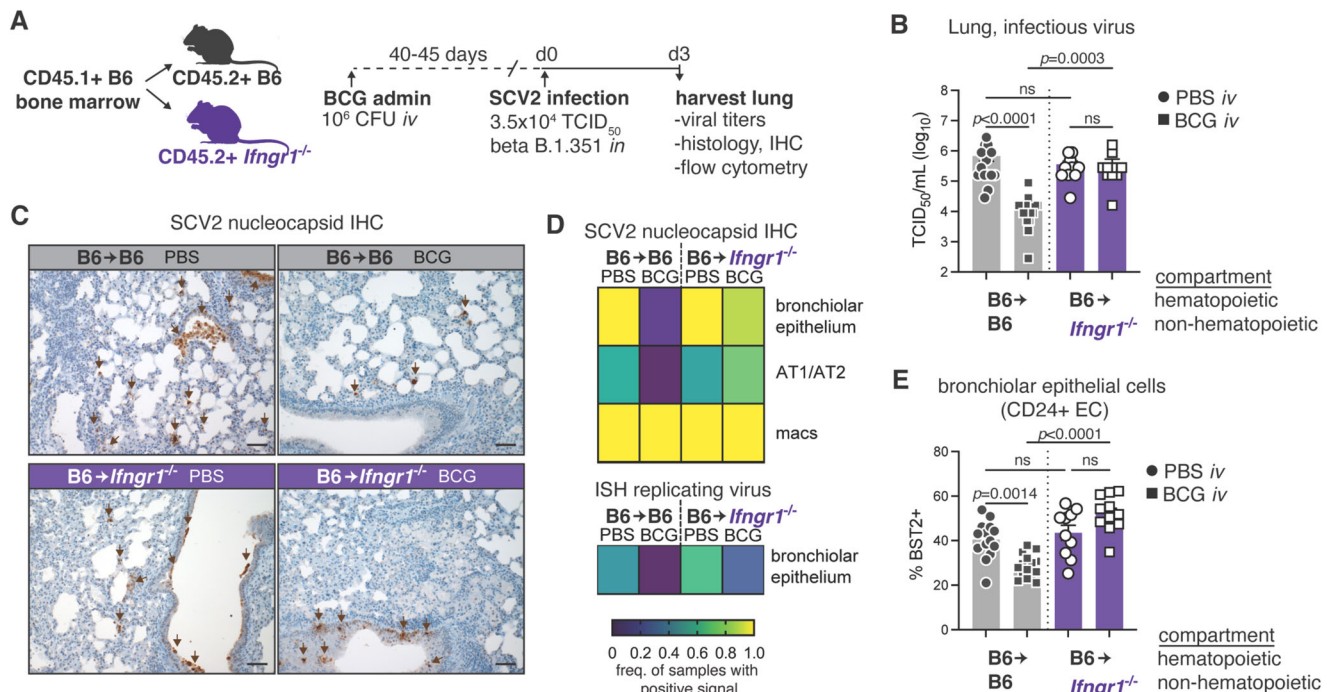

**Fig. 4 | Restriction of IFNγR1 signaling to the non-hematopoietic compartment is sufficient for *iv* BCG-induced protection against SARS-CoV-2 infection.** B6 or *Ifngr1*$^{-/-}$ CD45.2+ mice were irradiated and reconstituted with B6 congenic CD45.1+ bone marrow. Chimeras were inoculated with BCG or PBS *iv* 40–45 days prior to intranasal challenge with SARS-CoV-2 (SCV2) B.1.351. Lungs were harvested 3 days after viral challenge. **A** Schematic of experimental protocol. **B** Viral titers in lung homogenate from B6 or *Ifngr1*$^{-/-}$ chimeras as measured by TCID$_{50}$ assay (B6 PBS $n = 14$, B6 BCG $n = 15$, *Ifngr1*$^{-/-}$ PBS $n = 10$, *Ifngr1*$^{-/-}$ BCG $n = 11$; pooled from three independent experiments; Kruskal–Wallis with Dunn's post test). Not significant (ns) $p > 0.05$; ***$p < 0.001$; ****$p < 0.0001$. Bar graphs show geometric mean ± SEM.

**C** Representative lung histology images of SARS-CoV-2 nucleocapsid immunohistochemical staining. Scale bar = 50 μm. **D** Heat map representation of SARS-CoV-2 positivity across different cell types (IHC and ISH) as assessed by a study-blinded veterinary pathologist (B6 PBS $n = 9$, B6 BCG $n = 10$, *Ifngr1*$^{-/-}$ PBS $n = 6$, *Ifngr1*$^{-/-}$ BCG $n = 8$; pooled from two independent experiments). **E** Expression of BST2 by CD24+ epithelial cell populations as determined by flow cytometry (B6 PBS $n = 14$, B6 BCG $n = 15$, *Ifngr1*$^{-/-}$ PBS $n = 10$, *Ifngr1*$^{-/-}$ BCG $n = 11$; pooled from three independent experiments; one-way ANOVA with Tukey post test). Not significant (ns) $p > 0.05$. Bar graphs show mean ± SEM. Source data are provided as a Source Data file.

## IFNγ receptor signaling in non-hematopoietic cells is sufficient for *iv* BCG-induced protection against SARS-CoV-2

Given that SARS-CoV-2 primarily infects pulmonary epithelial cells (EC) and that IFNγ neutralization impacts the expression of anti-viral ISGs within the CD45-negative cellular compartment, we next wanted to determine whether IFNγ receptor signaling in the non-hematopoietic compartment is sufficient for *iv* BCG-driven control of viral loads. To do this, *Ifngr1*$^{-/-}$ mice were lethally irradiated and reconstituted with bone marrow cells from WT B6 congenic donors, so that all radio-sensitive immune cells could signal through the IFNγ receptor while all radioresistant cells, including the epithelial compartment, were *Ifngr1* deficient (B6→*Ifngr1*$^{-/-}$). Due to the essential role hematopoietic IFNγR1 signaling plays in controlling BCG bacterial loads[51], we did not generate reciprocal chimeras for these experiments. Rather, we generated WT B6 control chimeras (B6→B6) to account for radiation-induced stress and bone marrow reconstitution. Chimeras were injected *iv* with PBS or BCG and then challenged with SARS-CoV-2 after 40–45 days (Fig. 4A). As expected, lack of the IFNγ receptor on non-hematopoietic cells had no impact on SARS-CoV-2 infection in the absence of BCG (Fig. 4B). Viral loads were significantly lower in BCG inoculated control chimeras, but this protection was lost if the non-hematopoietic compartment was deficient in the IFNγ receptor indicating that BCG-induced IFNγ mediates control of viral loads through actions on non-hematopoietic cells (Fig. 4B).

To more specifically characterize the impact of IFNγ on SARS-CoV-2 infectivity of different cell types, we stained lung tissue sections for the SARS-COV-2 nucleocapsid and used in situ hybridization (ISH)

targeting replicating viral RNA to identify actively infected cells (Fig. 4C and Fig. S5A, B). We found that bronchiolar EC, pneumocytes, and macrophages were the major cell types immunoreactive for the SARS-CoV-2 nucleocapsid, with the bronchiolar epithelium the only lineage identified with active SARS-CoV-2 infection by positive probe signal (ISH) at the experimental endpoint (Fig. 4D and Fig. S5B). The immunoreactivity of macrophages is likely due to their role in their efferocytosis of debris of infected EC, rather than active infection[52,53], explaining the high occurrence of positive signal observed in *iv* BCG inoculated control chimeras despite the low viral titer enumerated by TCID$_{50}$ assay (Fig. 4C, D). Importantly, lack of the IFNγ receptor on non-hematopoietic cells was associated with increased immunoreactivity for SARS-CoV-2 nucleocapsid in bronchiolar EC and pneumocytes, as well as a higher frequency of rare ISH+ bronchiolar epithelial cells in *iv* BCG inoculated mice (Fig. 4C, D). Consistent with these findings, flow cytometric analysis showed the IFN-inducible anti-viral response protein BST2 was more highly expressed by *Ifngr1*$^{-/-}$ bronchial/bronchiolar EC (referred to as CD24 + EC) than *Ifngr1*$^{+/+}$ cells in *iv* BCG inoculated mice (Fig. 4E), indicative of higher viral load. In contrast, no differences were observed in BST2 expression by type-1 (AT1, CD326 + CD24-Pdpn + ) or type-2 (AT2, CD326 + CD24− MHCII+) pneumocytes when comparing B6 and *Ifngr1*$^{-/-}$ chimeras (Fig. S5C). Expression of CD274 (PDL1), which is strongly regulated by IFNγ, was significantly lower in all *Ifngr1*$^{-/-}$ epithelial cell types assessed from BCG inoculated animals confirming unresponsiveness of the epithelial compartment to bacteria-induced IFNγ (Fig. S5D). Together, these data support the conclusion that IFNγ produced following BCG injection controls SARS-CoV-2 infectivity and/or replication within the epithelial compartment.

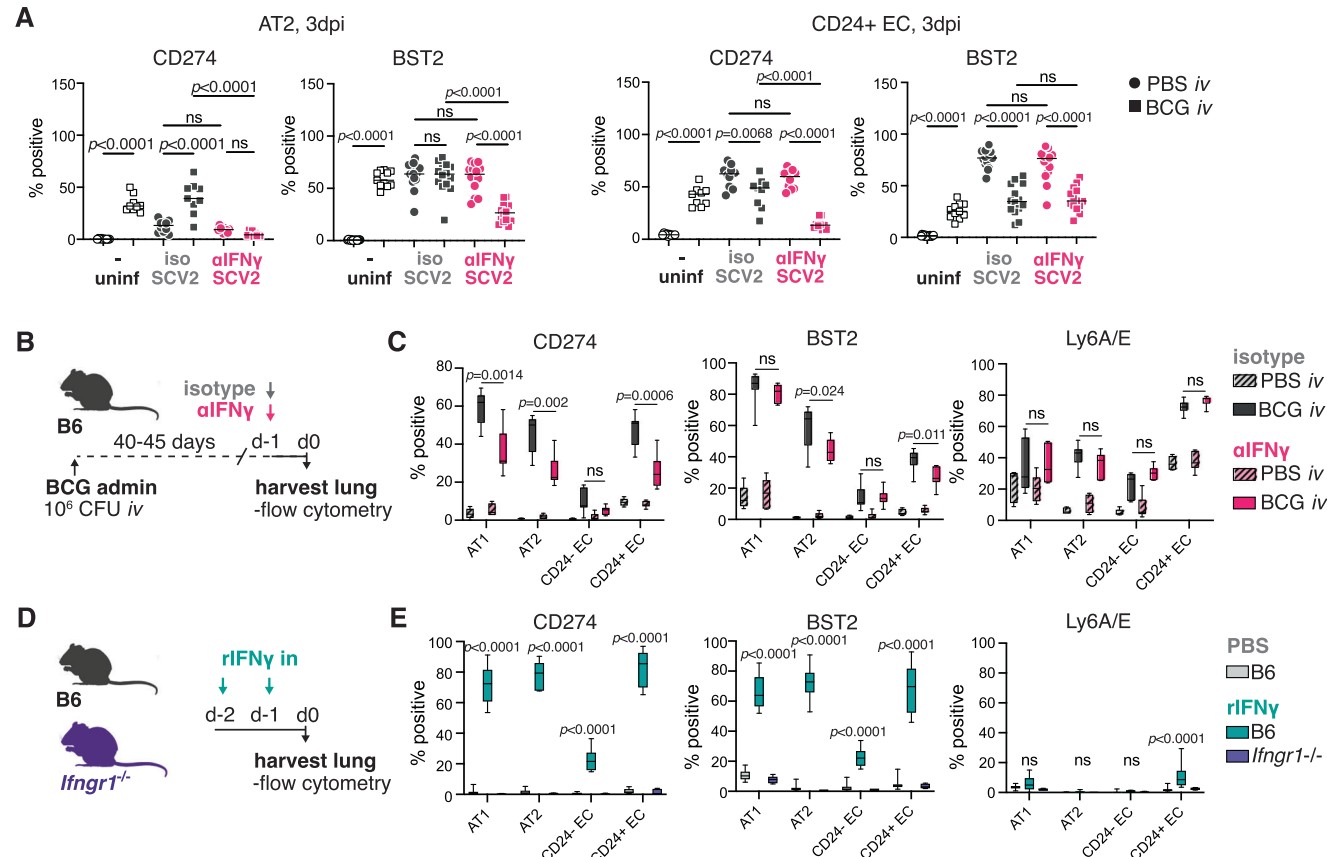

**Fig. 5 | IFNγ induces expression of anti-viral markers in pneumocytes and CD24+ epithelial cells. A** B6 mice were inoculated with BCG or PBS *iv* 40–45 days prior to intranasal challenge with SARS-CoV-2 (SCV2) B.1.351. Infected animals received an IFNγ neutralizing antibody or isotype control 1 day prior to and 1 day following SARS-CoV-2 instillation. Lungs were harvested 3 days after viral challenge and the indicated epithelial cell types were assessed for CD274 (*n* = 10/PBS group, *n* = 9/BCG group; pooled from 2 independent experiments; One-Way ANOVA with Tukey post-test) and BST2 (PBS *n* = 15, BCG *n* = 12, PBS + SCV2+iso *n* = 15, BCG + SCV2+iso *n* = 14, PBS + SCV2 + αIFNγ *n* = 14, BCG + SCV2 + αIFNγ *n* = 14; pooled from three independent experiments; one-way ANOVA with Tukey post-test) expression by flow cytometry. Not significant (ns) *p* > 0.05. **B, C** B6 mice were inoculated with BCG or PBS *iv* 40–45 days prior to receiving an IFNγ-neutralizing antibody or

isotype control. Lungs were harvested 1 day after anti-IFNγ treatment. **B** Schematic of experimental outline. **C** Expression of CD274, BST2, and Ly6A/E across different epithelial cell types (*n* = 7/BCG group; pooled from two independent experiments; two-tailed unpaired *t*-test between BCG isotype and BCG αIFNγ for each cell-type). Not significant (ns) *p* > 0.05. Data are displayed as median, quartiles ± range. **D, E** B6 or *Ifngr1*[−/−] mice were treated with PBS or rIFNγ intranasally on 2 consecutive days. Lungs were harvested 1 day after the last treatment. **D** Schematic of the experimental protocol. **E** Expression of CD274, BST2, and Ly6A/E across different epithelial cell types (B6 *n* = 10/group; two independent experiments; two-tailed unpaired *t*-test between PBS and rIFNγ treated B6 mice for each cell type). Not significant (ns) *p* > 0.05. Data are displayed as median, quartiles ± range. Source data are provided as a Source Data file.

## IFNγ promotes expression of anti-viral proteins in pneumocytes and bronchiolar epithelial cells

To gain a deeper understanding of epithelial responses to BCG and SARS-CoV-2, we assessed pulmonary EC by flow cytometry from control or *iv* BCG inoculated mice prior to or following SARS-CoV-2 challenge (gating shown in Fig. S6A). We focused on expression of IFN-inducible proteins with previously characterized roles in immune regulation (CD274, Ly6A/E)[54,55] and anti-viral activity (BST2)[48,56,57]. Interestingly, the expression pattern for each of the assessed proteins was distinct across different EC lineages (Fig. 5A and Fig. S6B, C). BCG inoculation and SARS-CoV-2 infection both drive strong upregulation of IFN responsive proteins in AT1 and AT2 cells, with the anti-viral protein BST2 induced to similar levels by BCG and SARS-CoV-2. CD24 + EC (bronchial/bronchiolar EC) also respond with upregulation of the assayed proteins following BCG and SARS-CoV-2, although their response to SARS-CoV-2 was much more pronounced than to BCG (Fig. 5A and Fig. S6B, C). We identified a small subset of CD326 + CD24− Pdpn- MHCII- cells (referred to as CD24− EC) in our analysis although these cells showed minimal responsiveness to BCG (Fig. S6B, C).

Given the ability of BCG to induce expression of proteins involved in anti-SARS-CoV-2 activity, we hypothesized that

mycobacterial-induced IFNγ may be inducing an "anti-viral" state in pulmonary EC prior to SARS-CoV-2 challenge thus limiting the infectivity of the virus. To examine this possibility, we inoculated animals with BCG and then neutralized IFNγ after 40–45 days before assessing the pulmonary EC 1 day later (ie. at the time we would usually challenge with SARS-CoV-2) (Fig. 5B). Again, we observed distinct expression patterns for CD274, BST2 and Ly6A/E across the different epithelial subsets. CD274 expression was the most strongly impacted by IFNγ neutralization, which was not unexpected given its well documented regulation by IFNγ[58]. Ly6A/E expression was not impacted by anti-IFNγ treatment, but BST2 was significantly reduced, albeit modestly, in AT2 and CD24 + EC, the two major cell types that are targeted for infection by SARS-CoV-2 (Fig. 5C). We next performed an experiment administering recombinant (r)IFNγ intranasally to naïve WT mice or animals that report expression of the IFN-inducible protein Irgm1 (M1Red) and then assessed EC responses by flow cytometry (Fig. 5D and Fig. S6D). These data closely match the observed epithelial response following *iv* BCG, with pneumocytes and CD24 + EC strongly upregulating CD274, BST2 and Irgm1 after rIFNγ treatment and only showing low-level expression of Ly6A/E (Fig. 5E and Fig. S6E, F). As observed with BCG, CD24- EC had a lower

response to IFNγ compared to the other EC subsets (Fig. 5E), except for Irgm1 which was similarly induced across all cell types following rIFNγ treatment (Fig. S6E, F). Together, our data demonstrate that BCG-driven IFNγ induces expression of IFN-regulated proteins by pulmonary EC, including the viral restriction factor BST2.

## Intranasal administration of recombinant IFNγ confers strong protection against SARS-CoV-2 in two mouse models

To test whether IFNγ is sufficient to confer protection against SARS-CoV-2 in the absence of BCG inoculation, we administered the recombinant cytokine intranasally to WT or $Ifngr1^{-/-}$ animals on the two days preceding and the day following SARS-CoV-2 challenge (Fig. 6A). At 3 days post infection, viral titers in the lung tissue were significantly lower in WT B6 animals that received intranasal rIFNγ, with more than 50% of treated animals having no detectable infectious particles at this timepoint (Fig. 6B). Importantly, no protection was observed in rIFNγ-treated $Ifngr1^{-/-}$ mice confirming that rIFNγ exerted anti-viral activities through its receptor (Fig. 6B). Immunohistochemical analysis of the SARS-CoV-2 nucleocapsid in lung sections showed a very similar picture, with reduced viral immunoreactivity in lungs from rIFNγ-treated WT B6 mice across all cell types assessed (Fig. 6C, D). Notably, intranasal rIFNγ also significantly protected against SARS-CoV-2-induced pulmonary inflammation and pneumonia in this mild infection model (Fig. 6C, E), indicating that it may also be protective against severe disease as manifested by marked tissue changes and lung damage[47].

To address the latter hypothesis, we performed an additional experiment in which we treated highly susceptible K18-hACE2 mice intranasally with rIFNγ prior to challenging with SARS-CoV-2 (rIFNγ d-2) and then assessed survival, body weight and viral titers (Fig. 6F). Based on our analysis of EC (Fig. 5), we hypothesized that IFNγ predominantly exerts its protective effects through induction of anti-viral programs in cells targeted for SARS-CoV-2 infection prior to viral exposure. To examine this possibility, we also included a group of mice in the survival study that started rIFNγ treatment at the peak of viral replication, 2 days after SARS-CoV-2 challenge (rIFNγ d+2) (Fig. 6F). Animals that received rIFNγ from d+2 and the PBS control group rapidly lost weight following SARS-CoV-2 exposure (Fig. 6G), with all mice succumbing to infection by 11 days post challenge (Fig. 6H). In contrast, rIFNγ treatment starting prior to SARS-CoV-2 protected against weight loss in the majority of animals and significantly improved survival (Fig. 6G, H), suggesting that IFNγ-induced responses are required at-or-near the time of viral exposure to prevent the establishment of SARS-CoV-2 infection and protect against disease. Viral titers assayed in whole lung homogenate at 5 dpi showed lower SARS-CoV-2 burdens with rIFNγ treatment, consistent with our findings in the B6 model (Fig. 6I). Finally, we assessed viral loads in individual cell types sorted from SARS-CoV-2-infected K18-hACE2 mice 2 days after challenge to determine whether rIFNγ preferentially affected anti-viral activity in a particular cell-type (sorting strategy shown in Fig. S7). As expected, EC subsets (AT1, AT2, and CD24 + EC) had the highest number of viral copies per µg of RNA, suggestive of active infection (Fig. 6J). SARS-CoV-2 copies were also detected at low levels in sorted macrophages and T cells, which likely resulted from "sticky" viral material released from surrounding dead/dying infected cells. In all cell types assessed, rIFNγ treatment reduced the number of viral copies recovered; however, this did not reach statistical significance in the AT2 group. Overall, these data demonstrate that pre-existing IFNγ responses limit SARS-CoV-2 infection and/or replication across several epithelial cell lineages thus protecting the host from virus-induced tissue damage and immunopathology.

## Discussion

Rapid and robust type-1 or type-3 IFN responses are crucial for effective control of viruses. One feature that makes SARS-CoV-2 such a successful pathogen is its ability to suppress host IFN responses and MHCI expression, much more so than other common respiratory viruses such as influenza A[59–62]. It is this immunosuppressive property that potentially makes respiratory viruses, such as SARS-CoV-2, amenable to restriction by pre-established IFN responses driven by concurrent or recent pulmonary infections. We have previously reported that pulmonary infection with BCG achieved through $iv$ inoculation provides striking protection against SARS-CoV-2-driven pathology and lethality, whereas subcutaneous administration does not[33]. In the present study we investigated the mechanism underlying this BCG-induced anti-viral resistance and formally demonstrate that IFNγ (type-2 IFN) signaling, specifically in non-hematopoietic cells, is essential for the observed protection against SARS-CoV-2. We further demonstrate that intranasal treatment with the recombinant cytokine restricts SARS-CoV-2 infection/replication and can protect the host from lethal disease.

Our findings reveal that, similar to type-1 and type-3 IFNs, pre-existing IFNγ can directly control SARS-CoV-2 viral loads; however, IFNγ does not appear to play a role during the natural course of SARS-CoV-2 infection since as shown here animals deficient in IFNγ receptor signaling do not display impaired viral control in the absence of BCG. This conclusion is in agreement with previous findings in both mice[25,63] and rhesus macaques[64]. Furthermore, while deficiencies in IFNγ signaling have been characterized in humans and linked with increased susceptibility to some viral pathogens[65], there is little evidence associating such mutations with increased risk of COVID-19 despite intensive investigation in this area. Rather, inborn errors in the IFN-I pathway are strongly associated with severe COVID-19[11,12] as well as other viral infections[66], whereas IFNγ signaling deficiencies are often linked with susceptibility to mycobacterial disease[67,68]. Redundant functions by type-1 and type-3 IFNs that are induced earlier during the course of viral infection are a potential explanation for lack of IFNγ requirement in these settings. Indeed, combined IFN receptor deficiencies (e.g., $Ifngr1$ and $Ifnar1$) render mice highly susceptible to SARS-CoV-2[25,69]. Furthermore, in the case of influenza, IFNγ responses have been linked to protection from secondary viral challenge as primary infection induced tissue-resident memory T cells are able to provide IFNγ rapidly upon re-exposure[18].

The above discussion highlights the importance of prompt IFNγ induction relative to the timing of viral exposure for beneficial outcomes. We directly tested this requirement in our study by comparing survival between animals treated with rIFNγ 2 days prior to versus 2 days following SARS-CoV-2 infection and found only pre-existing IFNγ responses could protect mice. Similar conclusions have been made in other animal studies assessing prophylactic versus therapeutic administration of recombinant type-1 or type-3 IFNs[25,30] or a type-1 IFN-inducing RIG-I agonist[31]. Interestingly, therapeutic treatment with IFNλ starting less than 24 hours following SARS-CoV-2 infection provided some protection in susceptible aged BALB/c or $Ifnar^{-/-}$ mice[25,70] that was significantly enhanced if the IFNλ treatment was combined with rIFNγ[25]. This finding suggests that combination treatments that include IFNγ could be effective in a therapeutic setting. In this regard, Beer et al., speculate that in their aged mouse model, IFNλ provides anti-viral signals whereas IFNγ restores an age-related delay in immune cell recruitment, which may explain why there was limited efficacy when each cytokine was administered individually. Our data suggest that IFNγ can also contribute directly to viral load control, particularly if it is delivered intranasally (as in the current study) versus subcutaneously[25].

The presence of an IFN response within the pulmonary compartment at the time of SARS-CoV-2 exposure is most likely to occur in the context of an ongoing or recent viral or bacterial infection. We show here that pulmonary BCG infection achieved through $iv$ administration induces a strong IFNγ signature in both myeloid and epithelial cells and that blockade of this response reverses protection against

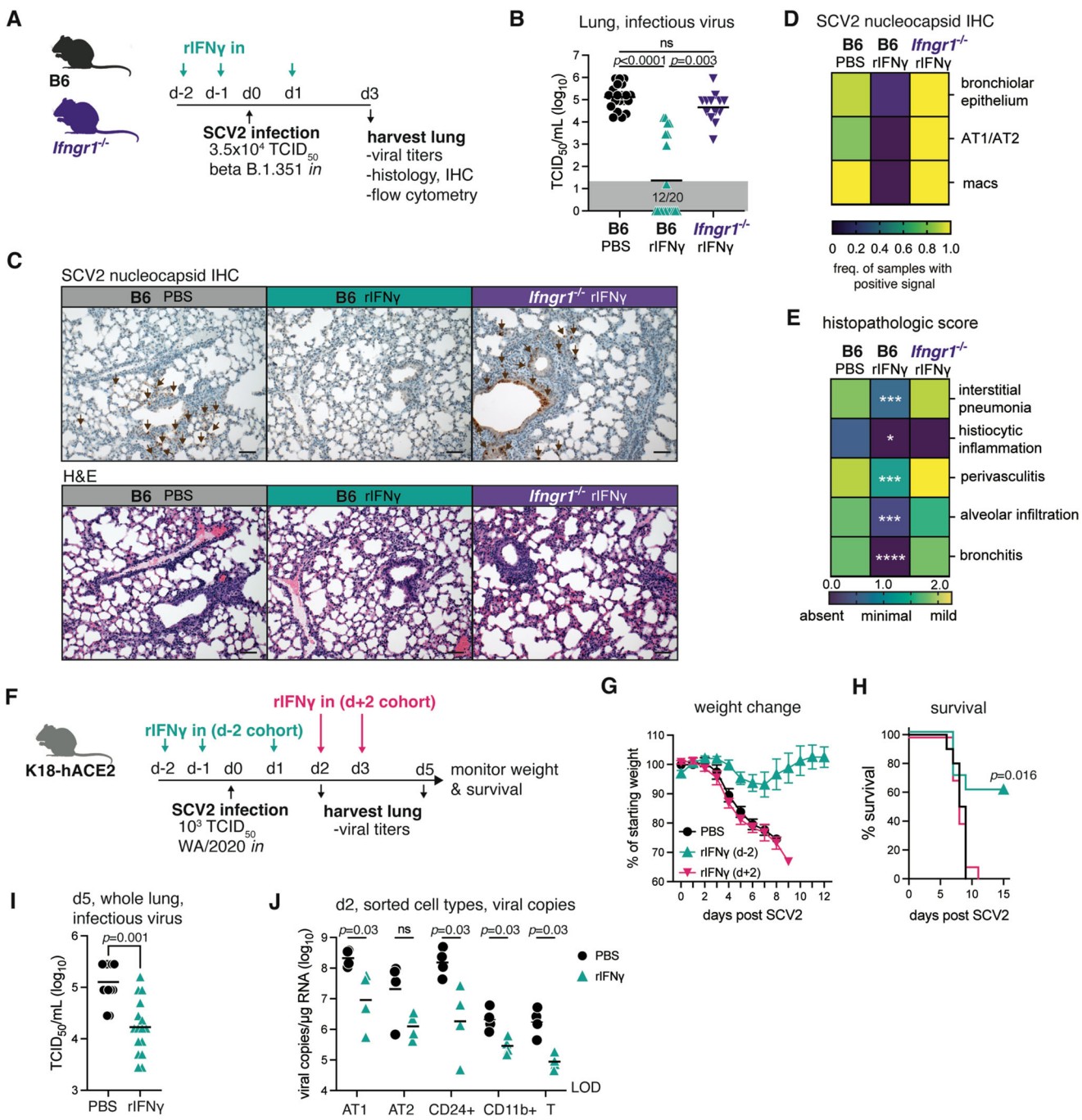

**Fig. 6 | Intranasal administration of recombinant IFNγ prior to viral challenge confers strong protection against SARS-CoV-2.** **A**–**E** B6 or *Ifngr1*⁻/⁻ mice were infected with SARS-CoV-2 (SCV2) B.1.351 and lungs harvested for analysis 3 dpi. Animals were treated with PBS or rIFNγ intranasally on days −2, −1, and 1 relative to viral challenge. **A** Schematic of experimental protocol. **B** Viral titers in lung homogenate as measured by TCID$_{50}$ assay (B6 $n$ = 20/group; pooled from four independent experiments. *Ifngr1*⁻/⁻ $n$ = 13; pooled from three independent experiments. Kruskal-Wallis with Dunn's post-test). Not significant (ns) $p$ > 0.05. Gray boxes denote values below limit of detection. **C** Representative lung histology images stained for SARS-CoV-2 nucleocapsid (upper panel) or H&E (lower panel) from two independent experiments. Scale bar = 50 μm. **D**, **E** Heat map representation of SARS-CoV-2 positivity across different cell types (**D**) or histopathologic score (**E**) as assessed by a study-blinded veterinary pathologist (B6 $n$ = 10/group; pooled from two independent experiments; two-tailed Mann–Whitney test between B6 PBS and B6 rIFNγ groups). Interstitial pneumonia $p$ = 0.0003,

histiocytic inflammation $p$ = 0.011, perivasculitis $p$ = 0.0007, alveolar infiltration $p$ = 0.0004, bronchitis $p$ < 0.0001. **F**–**J** K18-hACE2 mice were infected with SARS-CoV-2 WA/2020. Animals received intranasal PBS or rIFNγ at the indicated time points. **F** Schematic of experimental protocol. **G** Percent of starting weight (mean ± SEM) and **H** survival over time following viral challenge ($n$ = 10 mice/group; pooled from two independent experiments; Mantel–Cox test). **I** Viral titer in lung homogenate 5 dpi as measured by TCID$_{50}$ assay (PBS $n$ = 13, rIFNγ $n$ = 16; pooled from two independent experiments; two-tailed Mann–Whitney test). **J** Single-cell suspensions were prepared from lungs at 2 dpi. Cells were pooled into two replicates/experimental treatment groups each containing cells from 2–3 mice. The indicated cell types were sorted as per the gating strategy in Fig. S7. RNA was directly extracted, and viral copies were measured by PCR ($n$ = 4/group; pooled from two independent experiments; two-tailed Mann–Whitney test). Not significant (ns) $p$ > 0.05. LOD = level of detection. Source data are provided as a Source Data file.

SARS-CoV-2. Our scRNAseq and flow cytometry data demonstrate that *iv* BCG has a pronounced impact on the transcriptional landscape of pulmonary immune cells, with the replacement of resident alveolar macrophages by IFNγ primed monocyte-derived cells and the recruitment of Th1 cells being the most prominent changes. IFNγ transcript and protein were highly expressed by Th1 cells and *iv* BCG mice that lacked T cells were unable to control SARS-CoV-2 infection as effectively as their WT counterparts. T-cell deficiency had no significant impact on viral titers at 3 dpi in the absence of BCG, suggesting that the effect observed was due to BCG-induced T cell derived IFNγ. These findings are supported by a recent study where the authors found CD4+ T cells are central to *iv* BCG-induced protection against SARS-CoV-2[36]. Another interesting note from this experiment is that despite carrying a substantial bacterial load, *Tcra*[−/−] and *Ifngr1*[−/−] mice are still susceptible to SARS-CoV-2 suggesting that the mere presence of an ongoing BCG infection is insufficient to limit viral infection. This observation could explain the discrepancy in the lack of *iv* BCG-induced resistance in SARS-CoV-2-infected rodents reported by Kaufmann et al.[71] and the protection observed by us and other investigators[33–36] since failure to induce a sufficient IFNγ response following *iv* BCG administration due to potential differences in bacterial strain, preparation or dosing would not result in protection.

Importantly, in our murine models IFNγ mediated protection against SARS-CoV-2 occurs within the non-hematopoietic compartment. Given that pulmonary epithelial cells (ECs) are the primary target for SARS-CoV-2 infection[3–5] and that IFNγR-deficient ECs in the presence of an IFNγR-sufficient immune compartment still have increased viral loads, we propose that bacteria-induced IFNγ likely mediates its anti-viral effects directly within EC. However, without cell-type specific deletion of the *Ifngr1*, we cannot rule out contribution of other radioresistant cells or that IFNγ may also act through hematopoietic cells. EC and other cells targeted by viruses combat infection through IFN-regulated induction of host-derived viral restriction factors that are among a broader group of IFN-stimulated genes (ISGs)[6]. While thousands of ISGs have been identified[16], the precise mechanisms of anti-viral activity have only been described for a few dozen, which are typically regulated by type-1 and type-3 IFNs[6]. One such protein, BST2 (Tetherin/CD317), is expressed on the cell surface and interferes with the release of viral particles, including SARS-CoV-2, from infected cells[48,56,57]. BST2 is strongly induced by type-1 IFN but is also reported to be up-regulated by IFNγ depending on the cell-type assessed[16,57,72]. We show here that IFNγ promotes BST2 expression by pulmonary ECs and that the level induced by BCG infection is equivalent to that induced on AT2 cells by SARS-CoV-2 exposure. These data align with previous observations that pulmonary EC are highly responsive to intravenous rIFNγ treatment and influenza induced IFNγ (as measured by Irgm1-DSRed expression)[73]. Furthermore, a recent study showed significant enrichment of "anti-viral" transcriptional programs in lung EC, especially AT2, from mice inoculated with BCG *iv*[36]. Together the above findings suggest that IFNγ can directly stimulate and induce expression of ISGs in pulmonary EC that may be involved in restricting early viral replication and identify BST2 as a candidate for further study into IFNγ conferred anti-viral activity.

The major role of IFNγ in mediating host protection against SARS-CoV-2 appears to be through controlling viral load, which leads to lower levels of virus-induced pathology and mortality. This finding is clear in the rIFNγ model where viral loads were lower, lung pathology was reduced, and animals had improved survival. Interestingly, our scRNAseq, flow cytometry and cytokine multiplex data from *iv* BCG animals infected with SARS-CoV-2 showed that despite the higher viral load following IFNγ neutralization, markers of inflammation including IL-6 and CCL2 production were still reduced, along with less accumulation of inflammatory monocytes in the lung tissue compared to PBS controls. These data are consistent with the results of a clinical study where BCG vaccination of healthcare personnel was shown to reduce the production of pro-inflammatory cytokines upon exposure to irradiated SARS-CoV-2 ex vivo[74]. Together these findings suggest that BCG can dampen virus-induced inflammation independently of bacterial-induced IFNγ. One potential explanation is that the sustained immune response to BCG initiates negative feedback pathways of pro-inflammatory transcription factors (e.g., NFκB) thereby limiting virus-induced production of NFκB regulated cytokines including IL-6 and CCL2[75]. Alternatively, the short window of IFNγ neutralization employed in these studies may be enough to impact viral loads but residual IFNγ imprinting of the epithelium and other cell types is sufficient to protect against inflammatory cytokine production and immune cell recruitment.

Overall, our findings indicate that bacterial infections that specifically induce IFNγ responses within the lung may restrict SARS-CoV-2 infection. BCG administered *iv* appears to be particularly effective in this regard due to bacterial persistence in the lung tissue and a sustained IFNγ response[76]. Indeed, over time, the level of anti-viral protection afforded by *iv* BCG decreases likely due to the reduction in bacterial load and levels of IFNγ[33]. This may also explain why subcutaneous or intradermal delivered BCG has largely failed to protect against SARS-CoV-2 in mice[33,77] and humans[78–85], respectively. In these cases, bacteria either persist at low levels[33,86] or start to be cleared after a couple of weeks[87] resulting in modest baseline IFNγ responses that are highly heterogenous in the absence of cognate re-stimulation[88]. Interactions with other bacterial pathogens such as *Streptococcus pneumoniae* (Spn) that result in acute infections with predominant Th17 responses are also not protective, with one mouse study showing that Spn colonized animals are more susceptible to SARS-CoV-2 infection[89]. Meanwhile, aerosol infection with *Mycobacterium tuberculosis* (Mtb) does protect mice against SARS-CoV-2[37,38] independent of type-1 IFN signaling[39]. Like BCG, Mtb invokes a strong IFNγ response within the lung that potentially plays a part in the observed anti-viral effect. The fact that escalating Mtb dose increasingly restricts SARS-CoV-2[39] further supports a possible role for IFNγ in the murine models of Mtb-mediated SARS-CoV-2 restriction. Despite also driving strong IFNγ responses in humans, Mtb-infected individuals are not protected from SARS-CoV-2 and if anything, appear to show increased COVID-19 disease[90,91]. The reasons behind the discrepancy between the mouse and human data is likely multifaceted and related to a number of epidemiological and socioeconomic factors in addition to biological ones. It is clear however that IFNγ does possess anti-viral properties in humans, with rIFNγ treatment of a human pulmonary epithelial cell line effective at limiting SARS-CoV-2 infection in vitro[62,92,93] and subcutaneous rIFNγ treatment of a small group of moderately ill COVID-19 patients associated with reduced time to hospital discharge[29]. The work presented here raises the possibility that prophylactic intranasal rIFNγ administration could protect exposed individuals against SARS-CoV-2 infection or perhaps enhance the efficacy of other IFN treatments (e.g., pegylated IFNλ).

## Methods
### Study design
The aim of this study was the elucidate the mechanism/s by which concurrent mycobacterial infection protects against SARS-CoV-2. Two mouse models of SARS-CoV-2 infection were employed in this study: (1) commercially available K18-hACE2 transgenic mice that exhibit high suseptibility to SARS-CoV-2 and severe pathology[47] and (2) non-transgenic mice, either wildtype or gene knockouts, that display a mild disease phenotype when infected with SARS-CoV-2 B.1.351[40–44]. Animals were randomly assigned to groups of 3–8 mice for each experiment. Data from all experiments were pooled prior to analysis to identify reproducible and statistically significant differences between experimental groups. The number of mice per group, the number of experimental replicates and the statistical tests employed are reported in the figure legends. All data points are biological replicates. No

animals were excluded from analysis except for technical failure of intranasal inoculation. Endpoint criteria for survival studies were pre-determined in line with animal welfare recommendations set by the NIAID Animal Care and Use Committee.

## Mice

C57BL/6 J (JAX664), B6(Cg)-Ifnar1tm1.2Ees/J (JAX28288) and B6.Cg-Tg(K18-ACE2)2Prlmn/J hemizygous (JAX34860) mice were purchased from The Jackson Laboratory (Bar Harbor, ME); B6.PL-Thy1a/CyJ (JAX406), B6.SJL-Ptprca Pepcb/BoyJ (JAX2014), B6.129S7-Ifngr1tm1Agt/J (JAX3288) and B6.129S2-Tcratm1Mom/J (JAX2115) mice were acquired from the NIAID Contract Facility at Taconic Farms; M1Red mice[73] were bred onsite at NIAID. Mice were housed under specific pathogen−free conditions (individually ventilated caging, ambient temperature $22 \pm 3\,°C$, humidity $50 \pm 20\%$, 12 h light/dark cycle) with *ad libitum* access to food and water. Animals were randomly assigned to sex- and age-matched experimental groups. All animal studies were conducted in AALAC−accredited Biosafety Level 2 and 3 facil-ities at the NIAID, National Institutes of Health (NIH) in accordance with protocols approved by the NIAID Animal Care and Use Committee.

To generate bone marrow chimeras, CD45.2 + B6 and *Ifngr1*$^{-/-}$ mice received to two doses of 500cGy gamma-radiation, with a three-hour rest period between exposures. The following day, $10^7$ bone marrow cells from CD45.1 + B6.SJL donors were administered by intravenous injection. Animals were maintained on antibiotic drinking water for 3 weeks and rested for a further 5 weeks before the com-mencement of experiments. Flow cytometry was performed on per-ipheral blood cells to confirm successful reconstitution.

## Virology

SARS-CoV-2 strains USA-WA1/2020 (BEI Resources) and RSA B.1.351 N501Y (BEI Resources) were propagated in Vero-TMPRSS2 cells (kindly provided by Dr. Jonathan Yewdell, NIAID). Vero-TMPRSS2 cells were maintained in DMEM medium supplemented with Glutamax, 10% FCS and 250 μg/ml Hygromycin B gold (InvivoGen). Virus stock production was performed under BSL-3 conditions using DMEM medium supple-mented with Glutamax and 2% FCS. At 48 h post inoculation, culture supernatant and cells were collected, clarified by centrifugation for 10 min at $4\,°C$. Supernatant was collected, aliquoted and frozen at $-80\,°C$. Viral titers were determined by $TCID_{50}$ assay in Vero E6 cells (ATCC, CRL-1586) using the Reed and Muench calculation method. Full genome sequencing was performed at the NIAID Genomic Core (Hamilton, MT).

## BCG

BCG Pasteur from the Trudeau Collection was originally obtained from Dr. Sheldon Morris (Food and Drug Administration, MD) and main-tained as laboratory stock by serial passage. BCG was propagated in 7H9 broth supplemented with 10% oleic acid-albumin-dextrose-catalase (OADC) enrichment media (BD Biosciences) until mid-log phase. Bacteria were harvested, washed thrice and frozen down in aliquots until use. Colony forming units were enumerated by culturing on 7H11 agar for 3 weeks at $37\,°C$.

## Infections and treatments

BCG was prepared in PBS containing 0.05% Tween-80. A dose of $10^6$ CFU/mouse in 100 μL was delivered by intravenous or subcutaneous injection. Control animals received the same volume of PBS with 0.05% Tween-80.

Recombinant murine IFNγ (R&D Systems) was reconstituted in PBS containing 0.01% normal mouse serum, aliquoted and frozen at $-80\,°C$ until use. Just prior to administration, aliquots were thawed and diluted in PBS. Mice were anesthetized by isoflurane inhalation and 1 μg rIFNγ in a volume of 35 μL was administered by intranasal instillation

on days −2, −1, and 1 as indicated in the text and figures. Control animals received 35 μL PBS intranasally.

Anti-IFNγ (XMG1.2) or rat IgG1 isotype control (HPRN) was admi-nistered by intraperitoneal injection on days −1 (750 μg) and 1 (250 μg) as indicated in the text and figures. In some experiments, K18-hACE2 mice received an additional 250 μg dose on day 3. Anti-IFNAR (MAR1-5A3) or mouse IgG1 isotype control (MOPC-21) was administered by intraperitoneal injection on day −1 (2 mg) as indicated in the related figure. Antibodies were stored at $4\,°C$ until use and diluted in PBS just prior to administration. All antibodies were from BioXCell.

SARS-CoV-2 infections were performed under BSL-3 containment. Animals were anesthetized by isoflurane inhalation and a dose of $10^3$ $TCID_{50}$/mouse SARS-CoV-2 WA/2020 or $3.5 \times 10^4$ $TCID_{50}$/mouse SARS-CoV-2 B.1.351 was administered by intranasal instillation. Following infection, mice were monitored daily for weight change and clinical signs of disease by a study-blinded observer.

## Determination of viral copies by quantitative PCR

RNA was extracted using the Direct-zol RNA Miniprep kit following the manufacturer's instructions. E gene gRNA was detected using the QuantiNova Probe RT-PCR Kit and protocol and primers (forward primer: 5′- ACAGGTACGTTAATAGTTAATAGCGT-3′, reverse primer: 5′-ATATTGCAGCAGTACGCACACA-3′) and probe (5′-FAM-ACACTAGC CATCCTTACTGCGCTTCG-3IABkFQ-3′) as previously described[94]. The standard curve for each PCR run was generated using the inactivated SARS-CoV-2 RNA obtained from BEI (NR-52347) to calculate the viral copy number in the samples.

## Determination of viral titers by $TCID_{50}$ assay

Viral titers from lung homogenate were determined by plating in tri-plicate on Vero E6 cells (kindly provided by Dr. Sonja Best, NIAID) using 10-fold serial dilutions. Plates were stained with crystal violet after 96 hours to assess cytopathic effect. Viral titers were determined using the Reed-Muench method[95].

## Determination of colony-forming units

Bacterial burdens were enumerated by plating serially diluted lung homogenate on 7H11 agar (Sigma Aldrich) supplemented with 0.5% glycerol (Sigma Aldrich) and 10% OADC. BCG colonies were counted after a 3-week incubation at $37\,°C$.

## Preparation of single-cell suspensions from lungs

Lung lobes were diced into small pieces and incubated in RPMI con-taining 0.33 mg/mL Liberase TL and 0.1 mg/mL DNase I (both from Sigma Aldrich) at $37\,°C$ for 45 minutes under agitation (200 rpm). Enzymatic activity was stopped by adding FCS. The digested lung was filtered through a 70 μm cell strainer and washed with RPMI. Red blood cells were lysed with the addition of ammonium-chloride-potassium buffer (Gibco) for 3 minutes at room temperature. Cells were then washed with RPMI supplemented with 10% FCS. Live cell numbers were enumerated using AOPI staining on a Cellometer Auto 2000 Cell Counter (Nexcelom).

## Flow cytometry

To label cells within the pulmonary vasculature for flow cytometric analysis, 2 μg anti-CD45 SB702 (30-F11; Invitrogen) was administered by intravenous injection 3 minutes prior to euthanasia.

Single-cell suspensions prepared from lungs were washed twice with PBS prior to incubating with Zombie UV™ Fixable Viability Dye (1:1000) and TruStain FcX™ (1:200, clone 93; both from BioLegend) for 15 minutes at room temperature. Cocktails of fluorescently conjugated antibodies diluted in PBS and 10% Brilliant Stain Buffer (BD) were then added directly to cells and incubated for a further 20 minutes at room temperature. Anti-CD11b BUV805 (1:200, clone M1/70) and anti-CD26 BUV737 (1:100, clone H194-112) were from BD OptiBuild. Anti-CD4

BUV805 (1:200, clone GK1.5), anti-CD24 BUV737 (1:100, clone M1/69), anti-CD44 BV510 (1:200, clone IM7), anti-CD45 BUV395 (1:200, clone 30-F11), anti-Siglec F PECF594 (1:800, clone E50-2440), anti-TCR-beta chain BUV737 (1:200, clone H57-597) and anti-TCR-gamma-delta PECF594 (1:200, clone GL3) were from BD Horizon. Anti-CD8-beta APC-Cy7 (1:200, clone 53-6.7) and anti-CD274 AF488 (PDL1, 1:100, clone MIH5) were from Invitrogen. Anti-CD11c BV650 (1:200, clone N418), anti-CD31 APC-Cy7 (1:100, clone 390), anti-CD49f PE/Dazzle594 (1:200, clone GoH3), anti-CD64 PECy7 (1:100, clone X54-5/7.1), anti-CD88 PerCPCy5.5 (1:100, clone 20/70), anti-CD90.2 BV785 (1:200, clone 30-H12), anti-CD104 PECy7 (1:200, clone 346-11 A), anti-CD317 AF647 (BST2, 1:200, clone 927), anti-CD326 BV650 (Epcam, 1:200 clone G8.8), anti-IA/IE AF700 (MHCII, 1:100, clone M5/114), anti-Ly6A/E BV605 (Sca-1, 1:400, clone D7), anti-Ly6C BV785 (1:200, clone HK1.4), anti-Ly6G BV510 (1:200, clone 1A8), anti-NK1.1 BV650 (1:200, clone PK136) and anti-podoplanin BV421 (1:100, clone 8.1.1) were from BioLegend.

Cells were incubated in eBioscience™ Transcription Factor Fixation and Permeabilization solution (Invitrogen) for 2–18 hours at 4 °C and stained with cocktails of fluorescently-labeled antibodies against intracellular antigens diluted in Permeabilization Buffer (Invitrogen) for 30 minutes at 4 °C. Anti-IFNγ PECy7 (1:200, XMG1.2) was from BioLegend. Anti-FoxP3 PerCPCy5.5 (1:100, FJK-16s) and anti-Tbet e660 (1:200, 4B10) were from Invitrogen.

To perform intracellular cytokine staining, single-cell suspensions were incubated in RPMI supplemented with 10% FCS containing 1× protein transport inhibitor cocktail (ThermoFisher) for 5 hours at 37 °C. Cells were then stained as above.

Compensation was set in each experiment using UltraComp eBeads™ (Invitrogen) and dead cells and doublets were excluded from analysis. All samples were collected on a FACSymphony A5 SORP™ flow cytometer (BD) and analyzed using FlowJo software (version 10, BD)[96].

## Cell sorting
Single-cell suspensions prepared from lungs were pooled from 2–3 mice into two replicates per experimental group. Cells were incubated in TruStain FcX™ (clone 93; BioLegend) diluted in PBS for 10 minutes at 4 °C and subsequently stained with fluorescently conjugated antibodies diluted in PBS and 10% FCS for a further 30 minutes at 4 °C. Anti-CD11b BV785 (1:100, clone M1/70), anti-CD24 PE (1:100, clone M1-69), anti-CD31 APC-Cy7 (1:100, clone 390), anti-CD88 PerCPCy5.5 (1:100, clone 20/70), anti-CD104 PECy7 (1:200, clone 346-11 A), anti-CD326 BV650 (Epcam, 1:200, clone G8.8), anti-IA/IE AF700 (MHCII, 1:200, clone M5/114), anti-Ly6G BV510 (1:100, clone 1A8), anti-podoplanin BV421 (1:100, clone 8.1.1) and anti-TCR-beta chain APC (1:100, clone H57-597) were from BioLegend. Anti-CD45 SB702 (1:100, clone 30-F11) and anti-F4/80 FITC (1:100, clone BM8) were from Invitrogen. Following staining, cells were washed twice with PBS containing 5% FCS and stored on ice until sorting. Propidium iodide (1:1000, Thermofisher) was added to samples just prior to sorting to exclude dead cells. Populations of interest were sorted under BSL-3 containment on a FACSAria™ III cell sorter (BD) fitted with a 100 µm nozzle into PBS containing 20% FCS. The gating strategy is shown in Fig. S7.

## Multiplex cytokine array and ELISA
Cytokines were assessed in lung homogenate using a ProcartaPlex Luminex kit (ThermoFisher) according to the manufacturers' instructions and measured using a MagPix Instrument (R&D Systems). Interferon-lambda-3 was measured by Duoset ELISA (R&D Systems). Total protein was determined by BCA Assay (ThermoFisher). Cytokine levels were standardized to total protein content.

## Single-cell RNA sequencing
Single-cell suspensions were prepared from lungs as described above. An equal number of cells were pooled from all mice in a group. 10,000 cells from each group were loaded on a 10X Genomics Next GEM chip and single-cell GEMs were generated on a 10X Chromium Controller. Subsequent steps to generate cDNA and sequencing libraries were performed following 10X Genomics' protocol. Libraries were pooled and sequenced using Illumina NextSeq 200 and NovaSeq 6000 as per 10X sequencing recommendations. All samples had a sequencing yield of more than 237 million reads per sample.

The sequenced data were processed using Cell Ranger version 6.1.2 to demultiplex the libraries. The reads were aligned to *Mus musculus* mm10 and SARS-CoV-2 (MN981442.1) genomes to generate count tables that were further analyzed and visualized using Seurat version 4.1.2[97], dplyr[98], and ggplot2[99], with minor corrections to made to the output display using ggpubr[100], and ggrepel[101]. DEGs were calculated using the Wilcoxon Rank Sum test with Bonferroni correction ($p$ value < 0.05, log2 fold change limit of 0.25, and the stipulation that genes must appear in at least 10% of cells in either cluster). Clusters were defined by manual expert classification, based on cluster-specific DEGs using the R Seurat::FindMarkers function. Total scRNAseq data were split into three groups (CD45neg, myeloid and lymphoid as per Fig. S1A) based on expert curation prior to downstream analyses and sub-clustering. Each subset had independent scaling, PCA, UMAP, neighbor, and cluster analysis performed with 30 PC, 30 dims, and resolutions of 0.4 and 0.6.

## Single-cell gene set enrichment analysis
A list of genes whose products have been experimentally validated to contribute to anti-viral activity against SARS-CoV-2 was manually curated from Martin-Sancho et al.[48]. and Pfaender et al.[49]. scGSEA was performed with the "anti-SARS-CoV-2" gene list (Supplementary Data 11) using the escape package for R (version 1.10.0)[102]. Module scores (reported as "enrichment scores" in text, figures, and legends) were computed using the Seurat::AddModuleScore function in Seurat (version 4.4)[97]. Statistical significance was calculated using a Wilcoxon Rank sum test.

## Histology
Tissues were fixed in 10% neutral buffered formalin for 48–72 hours and embedded in paraffin. Embedded tissues were sectioned at 5 µm and dried overnight at 42 °C prior to staining. Specific anti-SARS-CoV-2 immunoreactivity was detected using a SARS-CoV-2 nucleoprotein antibody U864YFA140-4/CB2093 NP-1 (1:1000, Genscript). The secondary antibody was the Vector Laboratories ImPress VR anti-rabbit IgG polymer (cat# MP-6401). The tissues were then processed for immunohistochemistry using the Discovery Ultra automated stainer (Ventana Medical Systems) with a ChromoMap DAB kit (Roche Tissue Diagnostics cat#760–159). Detection of SARS-CoV-2 viral RNA was performed using the RNAscope 2.5 VS assay (Advanced Cell Diagnostic Inc.) on the Ventana Discovery ULTRA as previously described[103] and in accordance with the manufacturer's instructions. Briefly, tissue sections were deparaffinized and pretreated with heat and protease before hybridization with the antisense probe RNAscope 2.5 VS prove-V-nCoV2019-S-sense (Advanced Cell Diagnostics Inc, cat# 845709). Tissue slides were evaluated blindly by a board-certified veterinary pathologist for histopathological score and the presence/absence of SARS-CoV-2 nucleoprotein in bronchiolar epithelial cells, pneumocytes, and macrophages. Cell types were determined based on location and morphology.

## Statistical analyses
In all cases, statistical analyses were performed on pooled data from 2–3 independent experiments, each with 3–8 mice per group. Details for each analysis performed are reported in the figure legends. All data points are shown on the graphs and no animals were excluded except due to technical failure as outlined in the Study Design.

$P$ values were determined by two-tailed Student's unpaired $t$ test or two-tailed Mann–Whitney test when comparing two groups, or by one-way ANOVA with Tukey's post-test or Kruskal–Wallis test with

Dunn's post-test when comparing three or more groups using Graph-Pad Prism software (v9). *P* values below 0.05 were considered statistically significant.

### Figure visualization
Figures were generated in Adobe Illustrator and R[104] incorporating images from Biorender.com.

### Reporting summary
Further information on research design is available in the Nature Portfolio Reporting Summary linked to this article.

## Data availability
Single-cell RNA sequencing data generated in this study has been deposited to the NCBI GEO database and is available under the Accession ID GSE236601. Source data are provided with this paper.

## Code availability
All code and associated parameters used for scRNAseq analyses in this study are available at: https://doi.org/10.5281/zenodo.10068024[105].

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

## Acknowledgements
We are grateful to Dr. Daniel Barber (NIAID) and Dr. Sonja Best (NIAID) for discussion, Dr. Christine Nelson (NIAID) for assistance with setting up SARS-CoV-2 inactivation protocols, and Dr. David Eccles (Malaghan Institute) for bioinformatics support. We also thank Virgilio Bundoc and Robert Thompson for technical assistance; the National Cancer Institute Genomics Core for single-cell RNA sequencing; Dr. Craig Martens and the RML Genomics Unit for viral sequencing; the NIAID Research Technologies Branch for assistance with flow cytometry and the NIAID animal care staff. K.L.H. was partially supported by a Rutherford Postdoctoral Fellowship from Te Apārangi Aotearoa/Royal Society of New Zealand and by an independent research organization grant (to the Malaghan Institute) from the Health Research Council of New Zealand. S.I.O. and O.L. were supported by an independent research organization grant (to the Malaghan Institute) from the Health Research Council of New Zealand. This research was funded by the Intramural Program of NIAID, NIH.

## Author contributions
Conceptualization: K.L.H., S.N., C.G.F., A.S. Methodology: K.L.H., S.N., C.S.C., P.J.B. Investigation: K.L.H., S.N., C.S.C., P.J.B., V.P., E.P.A., S.D.O., D.O., J.L., M.C. Resources: N.L.G., B.A.P.L., R.F.J. Data curation and analysis: K.L.H., S.N., C.S.C., V.P., S.I.O. Writing—original draft: K.L.H., A.S. Writing—review and editing: K.L.H., S.N., C.S.C., P.J.B., V.P., C.G.F., D.J., O.L., K.D.M.-B., A.S. Visualization: K.L.H., S.N., C.S.C., S.I.O. Supervision: K.D.M.-B., D.J., O.L., A.S. Funding acquisition: A.S.

## Competing interests
The authors declare no competing interests.

## Additional information

[1]Immunobiology Section, Laboratory of Parasitic Diseases, National Institute of Allergy and Infectious Diseases, National Institutes of Health, Bethesda, MD 20892, USA. [2]Malaghan Institute of Medical Research, Wellington 6012, New Zealand. [3]Rocky Mountain Veterinary Branch, National Institute of Allergy and Infectious Diseases, National Institutes of Health, Hamilton, MT 59840, USA. [4]Inflammation and Innate Immunity Unit, Laboratory of Clinical Immunology and Microbiology, National Institute of Allergy and Infectious Diseases, National Institutes of Health, Bethesda, MD 20892, USA. [5]Immunoparasitology Unit, Laboratory of Parasitic Diseases, National Institute of Allergy and Infectious Diseases, National Institutes of Health, Bethesda, MD 20892, USA. [6]Flow Cytometry Section, Research Technologies Branch, National Institute of Allergy and Infectious Diseases, National Institutes of Health, Bethesda, MD 20892, USA. [7]SARS-CoV2- Virology Core, Laboratory of Viral Diseases, National Institute of Allergy and Infectious Diseases, National Institutes of Health, Bethesda, MD 20892, USA. [8]Immunology and Host Defense Group, School of Medical Sciences, Faculty of Medicine and Health, The University of Sydney, Camperdown, NSW 2006, Australia. [9]Centenary Institute, The University of Sydney, Camperdown, NSW 2050, Australia. ✉e-mail: khilligan@malaghan.org.nz; asher@niaid.nih.gov

