## [Peer Review File · Nature Communications]

Bacterial-induced or passively administered interferon gamma conditions the lung for early control of SARS-CoV-2Reviewers' Comments:

Reviewer #1:

Remarks to the Author:

In the submitted manuscript, Hilligan et al. investigate the mechanism by which intravenous (IV) BCG confers protection against SARS-CoV-2 challenge in a murine model. Prior work in the field from this lab and others have described the protective effect of IV BCG but the direct mechanism of action has yet to be fully elucidated. In the manuscript the authors find that IV instillation of BCG generates a strong antiviral response in the respiratory mucosa due to the production of IFN- γ . In this manuscript advanced methodologies to determine the mechanism of BCG induced IFN- γ production are used including single cell RNA sequencing, bone marrow chimeras, genetic knockouts, and antibody-mediated ablation. Further the authors demonstrate that administration of recombinant IFN- γ into the airways is sufficient to protect against SARS-CoV-2 challenge in a non-BCG vaccinated, *Ifngr*^{-/-} mouse.

This findings of this paper are of high significance to the field. It contains a plethora of data which are well-presented, and the manuscript is written articulately.

Points to consider:

Major:

1. Figure 4. There is a mismatch in the description of the bone marrow chimeras between (a) the main results text, (b) the figure legend, and (c) methods section. A. Main text at line 206 "To do this, *Ifngr*^{-/-} mice were lethally irradiated and reconstituted with bone marrow cells from WT B6 congenic donors". B. Figure legend (lines 706-708) "B6 congenic CD45.1+ mice were irradiated and reconstituted with either B6 CD45.2+ or *Ifngr*^{-/-} CD45.2+ bone marrow". C. Methods Section (lines 448-453) "To generate bone marrow chimeras, CD45.2+ B6 and *Ifngr*^{-/-} mice received to two doses of 500cGy gamma-radiation, with a three-hour rest period between exposures"

The first quote insinuates that *Ifngr*^{-/-} mice were irradiated and received wildtype bone marrow. This makes sense in the context of the paper as the non-hematopoietic cells would be missing *Ifngr*, while the bone marrow derived cells would have *Ifngr*. The line in the figure legend reflects the opposite, and the line in the methods section described that both types of chimeras were generated. As it stands it is all very confusing. This in turn makes the figure labels in 4B and 4E "B6 -> B6 / B6 -> *Ifngr*^{-/-}" all the more confusing and difficult to interpret.

2. Missing Experiment with bone marrow chimeras:

Building from the above point on the description of the bone marrow chimeras, I feel it is necessary to be do and show a study in which wildtype B6 mice are irradiated and given *Ifngr*^{-/-} bone marrow with follow-on IV BCG vaccination and SCV2 challenge. If the authors' hypothesis that the source of IFN- γ signaling lies in the non-radiosensitive, non-hematopoietic cells, then these mice should be protected.

3. Instances where the text and the data in Figures do not match.

a. Line 133 and Fig S1F: "elevated IFN γ responses levels and Tbet+ Th1 cells were only observed after iv administration and not in SCV2-susceptible animals administered BCG by the sc route". The figure shows that sc administration does in fact elevate Th1 cells, just not to the level of IV.

b. Line 132 and Fig 1I: "Together these data show that iv BCG induces a T and NK cell driven IFN γ response in the lung". The data in figure 1I demonstrates that IFN- γ expression in NK cells is not significant, so I am unsure how the authors can make this claim.

c. Line 157 and Fig 2d: "This treatment failed to further reduce . . ." actually it showed ~0.5 log better protection.

d. Line 164 and Fig 2f-g "...but this protection was lost in animals treated with anti-IFN γ ". The difference between isotype and anti IFN γ treated mice is not significant in either figure, so I don't know how you can make this claim.

e. Line 231; Fig S4c. ". . . no differences were observed in CD317 expression by AT1 and AT2 . . ." but AT1 with BCG had statistically higher CD317

4. Missing Experiment investigating IFN- γ at the protein level.

The authors assert that BCG induces T and NK cell driven IFN- γ responses, however they only investigate this at the transcript level. Figure 1H looks at IFN- γ protein, but this is for whole lung homogenate, so the source of the IFN- γ cannot be determined. Figure S1F quantifies the number of Tbet+ cells, but does not confirm they are in fact secreting IFN- γ . Confirming that T cells are the source of IFN- γ would strengthen the impact and rigor of the publication. The authors utilize Tcra-/- mice to demonstrate the mechanism is T cell dependent. Though they see an impact on viral burden, they do not check the IFN- γ levels of the Tcra-/- mice. This might be tested readily on leftover BAL fluid or lung homogenate.

5. Additional discussion needed in the text.

- a. Figures 2C-D: Blocking IFN- γ and its receptor did not completely block BCG mediated protection so it is important to address potential alternative mechanisms of protection that are likely contributing.
- b. A paper by Nigel Curtis's Group "Neonatal BCG Vaccination Reduces Interferon- γ Responsiveness to Heterologous Pathogens in Infants From a Randomized Controlled Trial" (PMID: 31990350) found that BCG enhanced IFN- γ levels, but responses varied with heterologous pathogen. It is important to cite this paper in the discussion and evaluate what it could mean in the context of the authors' study.

Minor Points:

6. Lines 198-199. "BCG-induced IFN- γ acts on primary epithelial cells infected by SCV2" The authors should show evidence of ISG up-regulation in AT1, AT2, EC CD24+, and EC CD24- cells. They should already have the data in scRNAseq database.
7. Line 174 and Fig.3 Was histology performed to see if inflammation was reduced?
8. Line 189: Why is IP-10 excluded from the list?
9. Fig 1h is about mice with no SCV2 infection. Presumably, however, the rest of the Fig. 1 are WITH SCV2 challenge. It would be helpful to clarify this.
10. Typo: Line 284 "chages"
11. Typo: Fig 4c "bronciolar"
12. Typo: Line 196 "apparant"

Reviewer #2:

Remarks to the Author:

In the current study, the authors look at the effect of iv BCG pretreatment on disease following SARS-CoV-2 infection. They show that it is protective against viral load and disease and this is mediated by IFN γ acting on epithelial cells.

Major Comments:

1. Need to include data on the impact of IFN γ pretreatment before infection – do the mice get sick/ lose weight. The observations are interesting, but I think there are some challenges to this being used as a preventative approach, with a risk of non-specific damage to the lungs. It would seem to be more important for dissecting how IFN γ can protect, for example in the context of Th1 T cells.
2. Some commentary on why the effect is so long lived would be interesting. If the mice are treated with rifampicin to clear the BCG is it still protective?
3. I would like to see more mechanism of protection by IFN γ , Why were only 3 ISG looked at? How were they selected? In line 251 you say given the ability of BCG to induce proteins – but don't show this. Deeper analysis of the DEG from the seq data would be invaluable. Can you show that there are increased anti-viral genes and link to the ISG data later? E.g. CD317 is not in your gene list on the volcano plot.

Minor

1. Put the gene name (tetherin) for CD317 the first time it is mentioned (line 229) I think, that would make interpretation easier.
2. I think lines 311-315 are speculative. If you did the same model with influenza, I think you might see similar things.
3. Line 323 is not supported by the data – there is no data showing that IFN γ is replacing type 1 or 3 immunity. It is showing it can protect, but not that it replaces
4. In panels where it says No. DEG (e.g. 1E) it reads like no rather than number, can you rephrase
5. Figure 1I is very data dense and hard to interpret.
6. Figure 3B – not clear what significance is being compared to. Why doing fold change rather than actual values and show the control.
7. Panel 5C – put Y axis labels on all graphs

We would like to thank the Editor and Reviewers for their helpful feedback on our paper. In our revised manuscript we have addressed the Reviewers comments with the inclusion of new data, additional analyses and discussion. Please find our point-by-point response below.

REVIEWER COMMENTS

Reviewer #1 (Remarks to the Author):

In the submitted manuscript, Hilligan et al. investigate the mechanism by which intravenous (IV) BCG confers protection against SARS-CoV-2 challenge in a murine model. Prior work in the field from this lab and others have described the protective effect of IV BCG but the direct mechanism of action has yet to be fully elucidated. In the manuscript the authors find that IV instillation of BCG generates a strong antiviral response in the respiratory mucosa due to the production of IFN- γ . In this manuscript advanced methodologies to determine the mechanism of BCG induced IFN- γ production are used including single cell RNA sequencing, bone marrow chimeras, genetic knockouts, and antibody-mediated ablation. Further the authors demonstrate that administration of recombinant IFN- γ into the airways is sufficient to protect against SARS-CoV-2 challenge in a non-BCG vaccinated, *Ifngr*^{-/-} mouse.

This findings of this paper are of high significance to the field. It contains a plethora of data which are well-presented, and the manuscript is written articulately.

We thank the Reviewer for his/her positive assessment of our manuscript and for his/her helpful feedback. In response to the points raised we have now added new data and addressed several issues that needed clarification. These changes to the manuscript are highlighted here in yellow.

Points to consider:

Major:

1. Figure 4. There is a mismatch in the description of the bone marrow chimeras between (a) the main results text, (b) the figure legend, and (c) methods section. A. Main text at line 206 "To do this, *Ifngr*^{1-/-} mice were lethally irradiated and reconstituted with bone marrow cells from WT B6 congenic donors". B. Figure legend (lines 706-708) "B6 congenic CD45.1+ mice were irradiated and reconstituted with either B6 CD45.2+ or *Ifngr*^{1-/-} CD45.2+ bone marrow". C. Methods Section (lines 448-453) "To generate bone marrow chimeras, CD45.2+ B6 and *Ifngr*^{1-/-} mice received to two doses of 500cGy gamma-radiation, with a three-hour rest period between exposures". The first quote insinuates that *Ifngr*^{1-/-} mice were irradiated and received wildtype bone marrow. This makes sense in the context of the paper as the non-hematopoietic cells would be missing *Ifngr*¹, while the bone marrow derived cells would have *Ifngr*. The line in the figure legend reflects the opposite, and the line in the methods section described that both types of chimeras were generated. As it stands it is all very confusing. This in turn makes the figure labels in 4B and 4E "B6 -> B6 / B6 -> *Ifngr*^{1-/-}" all the more confusing and difficult to interpret.

*We apologise for the confusion. There was an error in the Figure Legend which has now been corrected. To clarify, there were two types of chimeras generated. One group were B6 CD45.2+ mice irradiated and reconstituted with B6 CD45.1+ bone marrow (denoted as B6->B6). The second group were *Ifngr*^{1-/-} CD45.2+ mice irradiated and reconstituted with B6 CD45.1+ bone marrow (denoted as B6->*Ifngr*^{1-/-}). We have now updated the text and the figure to make the distinction clearer, with haematopoietic or non-haematopoietic compartments noted alongside the graphs.*

“To do this, *lfng1*^{-/-} mice were lethally irradiated and reconstituted with bone marrow cells from WT B6 congenic donors, so that all radio-sensitive immune cells could signal through the IFN γ receptor while all radio-resistant cells, including the epithelial compartment, were *lfng1* deficient (B6->*lfng1*^{-/-}).”

2. Missing Experiment with bone marrow chimeras: Building from the above point on the description of the bone marrow chimeras, I feel it is necessary to be do and show a study in which wildtype B6 mice are irradiated and given *lfng1*^{-/-} bone marrow with follow-on IV BCG vaccination and SCV2 challenge. If the authors' hypothesis that the source of IFN- γ signaling lies in the non-radiosensitive, non-hematopoietic cells, then these mice should be protected.

*We agree that this would be an interesting experiment; however, we intentionally designed the experiment in this manner because we believe it would be confounded by the outgrowth of BCG in mice that lack hematopoietic *lfng1* (as is observed in complete knockout mice, FigS2). This is not an issue in *lfng1*^{-/-} mice reconstituted with B6 bone marrow (Rottman, 2008, Plos Med) (1) or with the short-term anti-IFN γ treatment (FigS2) and allows for a more fair comparison. Given that the non-hematopoietic *lfng1*^{-/-} phenocopied the whole-body knockout, we think that this adequately supports the conclusion that *lfng1* signaling is sufficient for the anti-viral protection afforded by BCG. At this stage we cannot rule out the possibility that *lfng1* signaling in the hematopoietic compartment could also contribute to anti-viral activity in the absence of *lfng1* in the non-hematopoietic compartment. Therefore, we have adjusted the wording in the text to move away from *lfng1* signaling being “required” in the non-hematopoietic compartment to being “sufficient” to account for this:*

Section header: “IFN γ receptor signaling in non-hematopoietic cells is sufficient for iv BCG induced protection against SCV2.”

Figure 4: “Restriction of IFN γ R1 signaling to the non-hematopoietic compartment is sufficient for iv BCG induced protection against SCV2 infection.”

We also explain our rationale for not including reciprocal chimeras in the text:

“Due to the essential role hematopoietic IFN γ R1 signaling plays in controlling BCG bacterial loads (1), we did not generate reciprocal chimeras for these experiments.”

Furthermore, generation of new chimeras would result in a substantial delay in the publication of our findings since in addition to the time involved in constructing the chimeras there have recently been unforeseen shutdowns of our animal BSL3 facility and it is unclear when we will again be in a position to perform SCV2 challenge infections. We hope the reviewer will sympathize with our situation and will find our proposed clarifications to the text as adequately addressing her/his point. We have discussed these problems with the editor and she has indicated that from her perspective the above revisions in principle should be a sufficient response to the issue raised.

3. Instances where the text and the data in Figures do not match.

a. Line 133 and Fig S1F: “elevated IFN γ responses levels and Tbet⁺ Th1 cells were only observed after iv administration and not in SCV2-susceptible animals administered BCG by the sc route”. The figure shows that sc administration does in fact elevate Th1 cells, just not to the level of IV.

We thank the reviewer for bringing this to our attention. The text has been modified as follows: “Importantly, IFN γ levels and the number of Tbet⁺ Th1 cells were significantly higher after iv administration compared to sc BCG inoculation, which we have previously shown is unable to protect animals against SCV2 (Fig1I and FigS1F) (2).”

b. Line 132 and Fig 1I: “Together these data show that iv BCG induces a T and NK cell driven IFN γ response in the lung”. The data in figure 1I demonstrates that IFN- γ expression in NK cells is not significant, so I am unsure how the authors can make this claim.

To clarify, the statistical analysis shown was comparing PBS+SCV2 to BCG+SCV2. While there was no statistical difference in *lfn* expression by NK cells between these two groups, the data did show that *lfn* was expressed by NK cells after iv BCG which is what was being referred to in the text. We agree that the violin plot representation was not clear. This has now been replaced with a UMAP plot in FigS1G with an *lfn* expression overlay to simply show that the NK cell cluster does express *lfn* transcripts.

To specifically show IFN γ protein expression in different cell types after BCG in the absence of SCV2 infection, we now include new data in Fig1I where we quantify the number of IFN γ ⁺ cells in the lung by flow cytometry. To capture a more accurate representation of cytokine production in vivo, these samples were incubated with protein transport inhibitors for 5 hours and were not re-stimulated ex vivo. These data show that iv BCG induces a higher number of total IFN γ ⁺ cells in the lung at 28dpi compared to sc BCG and that the majority of IFN γ ⁺ cells are CD4⁺ T cells (74 \pm 2%). The second most frequent population of IFN γ ⁺ cells are NK cells (8 \pm 1.5%) followed by CD8⁺ T cells (4.3 \pm 0.6%).

c. Line 157 and Fig 2d: “This treatment failed to further reduce . . .” actually it showed ~0.5 log better protection.

Reviewer Figure 1: viral loads at 3 days post SCV2 challenge.

d. Line 164 and Fig 2f-g “...but this protection was lost in animals treated with anti-IFN γ ”. The difference between isotype and anti-IFN γ treated mice is not significant in either figure, so I don’t know how you can make this claim.

This conclusion was made based on the fact that there is a significant difference between PBS and BCG in the isotype treated mice but there is no significant difference between PBS and BCG in the anti-IFN γ treated animals. Unfortunately, due to 2 outliers in the BCG iv + isotype group, statistical significance was not achieved between this group and the BCG iv + anti-

IFN γ ; however, there is a clear trend towards anti-IFN γ treated mice being more susceptible to SCV2 infection compared to isotype controls.

This has now been qualified in the text as follows: “iv BCG protected K18-hACE2 mice against weight loss and resulted in lower viral loads, but no significant protection was observed in animals treated with anti-IFN γ (Fig2F-G). While a direct comparison between isotype and anti-IFN γ treated mice inoculated with BCG prior to viral challenge did not reach statistical significance, animals treated with anti-IFN γ showed a clear trend towards more severe disease and higher viral loads (Fig2F-G).”

e. Line 231; Fig S4c. “. . . no differences were observed in CD317 expression by AT1 and AT2 . . .” but AT1 with BCG had statistically higher CD317.

This has now been clarified in the text as follows: “In contrast, no differences were observed in BST2 expression by type-1 (AT1, CD326+ CD24- Pdpn+) or type-2 (AT2, CD326+ CD24- MHCII+) pneumocytes when comparing B6 and Ifngr1-/- chimeras (FigS5C).”

4. Missing Experiment investigating IFN- γ at the protein level.

The authors assert that BCG induces T and NK cell driven IFN- γ responses, however they only investigate this at the transcript level. Figure 1H looks at IFN- γ protein, but this is for whole lung homogenate, so the source of the IFN- γ cannot be determined. Figure S1F quantifies the number of Tbet+ cells, but does not confirm they are in fact secreting IFN- γ . Confirming that T cells are the source of IFN- γ would strengthen the impact and rigor of the publication. The authors utilize Tcra-/- mice to demonstrate the mechanism is T cell dependent. Though they see an impact on viral burden, they do not check the IFN- γ levels of the Tcra-/- mice. This might be tested readily on leftover BAL fluid or lung homogenate.

We agree that additional information about the cellular source of IFN γ protein would strengthen the manuscript. To address this point, we now include new data in Fig11 which is described in response to point 3b above.

*The Reviewer suggests an interesting experiment of measuring IFN γ in lung homogenate of Tcra-/- to confirm our hypothesis that residual IFN γ signaling in these animals drives the small level of anti-viral activity observed. Indeed, there is previously published data demonstrating measurable IFN γ levels in the spleen of Tcrb-/- animals injected with BCG iv (Ladel et al 1995, EJI) (3). *This paper is now referenced in the text.**

5. Additional discussion needed in the text.

a. Figures 2C-D: Blocking IFN- γ and its receptor did not completely block BCG mediated protection so it is important to address potential alternative mechanisms of protection that are likely contributing.

We now discuss one likely candidate mechanism that chronic BCG infection may be regulating the activity of pro-inflammatory transcription factors, such as NF- κ B.

“...One potential explanation is that the sustained immune response to BCG initiates negative feedback pathways of pro-inflammatory transcription factors (eg. NF κ B) thereby limiting virus induced production of NF κ B regulated cytokines including IL-6 and CCL2 (4).”

b. A paper by Nigel Curtis's Group “Neonatal BCG Vaccination Reduces Interferon- γ

Responsiveness to Heterologous Pathogens in Infants From a Randomized Controlled Trial” (PMID: 31990350) found that BCG enhanced IFN- γ levels, but responses varied with heterologous pathogen. It is important to cite this paper in the discussion and evaluate what it could mean in the context of the authors’ study.

We thank the reviewer for bringing this to our attention. We have now added discussion of this study and other human BCG trials as follows:

“Interestingly, our scRNAseq, flow cytometry and cytokine multiplex data from iv BCG animals infected with SCV2 showed that despite the higher viral load following IFN γ neutralization, markers of inflammation including IL-6 and CCL2 production were still reduced, along with less accumulation of inflammatory monocytes in the lung tissue compared to PBS controls. These data are consistent with the results of a clinical study where BCG vaccination of healthcare personnel was shown to reduce the production of pro-inflammatory cytokines upon exposure to irradiated SCV2 ex vivo (5).” ...

“Overall, our findings indicate that bacterial infections that specifically induce IFN γ responses within the lung may restrict SCV2 infection. BCG administered iv appears to be so effective in this regard due to bacterial persistence in the lung tissue and sustained IFN γ response. Indeed, over time, the level of anti-viral protection afforded by iv BCG decreases likely due to the reduction in bacterial load and levels of IFN γ (2). This may also explain the why subcutaneous or intradermal delivered BCG has largely failed to protect against SCV2 in mice (2, 6) and humans (7-14), respectively. In these cases, bacteria either persist at low levels (2, 15) or start to be cleared after a couple of weeks (16) resulting in modest baseline IFN γ responses that are highly heterogenous in the absence of cognate re-stimulation (17).”

Minor Points:

6. Lines 198-199. “BCG-induced IFN- γ acts on primary epithelial cells infected by SCV2
“ The authors should show evidence of ISG up-regulation in AT1, AT2, EC CD24+, and EC CD24- cells. They should already have the data in scRNAseq database.

The scRNASeq data included in the manuscript was performed on samples isolated 3 days after SCV2 infection, and we therefore cannot assess how BCG influences ISG expression in the absence of SCV2 infection. To address this, we performed the flow cytometry experiments shown in Fig5, which provide evidence of surface protein expression on the epithelial cell types assessed.

In response to reviewer 2, point 3, we now include a new analysis of our scRNASeq data in FigS4B-C, where we use single cell gene set enrichment analysis (scGSEA) to show that anti-IFN γ treatment of iv BCG mice significantly reduces the enrichment of genes with validated anti-SCV2 activity in a number of different cell types despite viral loads being higher under these conditions. In contrast, no significant differences (except for B cells) were observed between PBS+SCV2+isotype versus PBS+SCV2+anti-IFN γ groups.

Further to this, we now also discuss a bioRxiv pre-print that became available after our submission where the Authors perform scRNASeq on lungs from control mice or animals inoculated iv with BCG 21 days prior (Lee et al, 2023, bioRxiv) (18). These data independently confirm our flow cytometry findings, showing that pulmonary epithelial cells express an anti-viral transcriptional signature after iv BCG. Together with our new analysis showing that IFN γ neutralization reduces the enrichment of anti-SCV2 genes, these data provide strong support for iv BCG induced IFN γ driving a broad anti-viral response in pulmonary epithelial cells.

7. Line 174 and Fig.3 Was histology performed to see if inflammation was reduced?

Histology was performed on these samples; however, as iv BCG itself induces considerable inflammation it is difficult to resolve differences between BCG and SCV2-driven inflammation in the B6 mouse model (see Fig4 for example).

Given this limitation, we unfortunately cannot draw any conclusions from histology samples about how IFN γ neutralization impacts specifically on SCV2-induced inflammation in mice previously inoculated with BCG.

8. Line 189: Why is IP-10 excluded from the list?

IP10 is now included.

9. Fig 1h is about mice with no SCV2 infection. Presumably, however, the rest of the Fig. 1 are WITH SCV2 challenge. It would be helpful to clarify this.

To make the distinction clearer, we have added +SCV2 to all the plots relating to the scRNASeq data.

10. Typo: Line 284 "chages **corrected**

11. Typo: Fig 4c "broncioloar" **corrected**

12. Typo: Line 196 "apparant" **corrected**

Thanks for pointing these out!

Reviewer #2 (Remarks to the Author):

In the current study, the authors look at the effect of iv BCG pretreatment on disease following SARS-CoV-2 infection. They show that it is protective against viral load and disease and this is mediated by IFN γ acting on epithelial cells.

We thank the Reviewer for her/his valuable feedback that has helped improve our manuscript. In response to the points raised we have now added new data and addressed several issues that needed clarification. These changes to the manuscript are highlighted here in yellow.

Major Comments:

1. Need to include data on the impact of IFN γ pretreatment before infection – do the mice get sick/ lose weight. The observations are interesting, but I think there are some challenges to this being used as a preventative approach, with a risk of non-specific damage to the lungs. It would seem to be more important for dissecting how IFN γ can protect, for example in the context of Th1 T cells.

In our experiments we did not observe any adverse effects of intranasal IFN γ pre-treatment in the short-term. WT B6 mice treated with rIFN γ did not lose any weight in the days following treatment or SCV2 infection. Furthermore, no signs of illness were noted by study-blinded observers using a clinical scoring system from 0-3, where 0=no signs of disease present, 1=mild disease (ruffled fur, reduced movement), 2=moderate disease (ruffled fur, reduced movement, dehydration, hunched) and 3=severe disease (moribund) (Reviewer Fig2).

Furthermore, assessment of lung histology slides from mice treated with rIFN γ and infected with SCV2 show minimal signs of inflammation or damage as determined by a board-certified veterinary pathologist (Fig6C, E). While these samples were collected post SCV2 infection, the lack of tissue damage suggests that rIFN γ treatment itself does not cause significant pathology in the short-term.

While these experiments support the absence of rIFN γ mediated pathology, we acknowledge that proper safety studies would be required before any clinical trial involving intranasal rIFN γ as an anti-viral to prevent SCV2 infection. We do note that rIFN γ is currently used in the clinic via subcutaneous administration without safety issues.

Reviewer Figure 2: weights and clinical scores of animals following rIFN γ and SCV2 infection.

2. Some commentary on why the effect is so long lived would be interesting. If the mice are treated with rifampicin to clear the BCG is it still protective?

This is an interesting question. While we have not done an experiment clearing BCG, we expect that protection would be lost as a pre-existing IFN γ response is key for the anti-viral effect observed (Fig6). We do show in our 2022 J Exp Med paper, that BCG protection against SCV2 wanes over time which we attribute to declining bacterial loads and the associated Th1 response. We have now added further mention of this in the Discussion as follows:

“Overall, our findings indicate that bacterial infections that specifically induce IFN γ responses within the lung may restrict SCV2 infection. BCG administered iv appears to be particularly effective in this regard due to bacterial persistence in the lung tissue and sustained IFN γ response (19). Indeed, over time, the level of anti-viral protection afforded by iv BCG decreases likely due to the reduction in bacterial load and levels of IFN γ (2).”

3. I would like to see more mechanism of protection by IFN γ , Why were only 3 ISG looked at? How were they selected? In line 251 you say given the ability of BCG to induce proteins – but don't show this. Deeper analysis of the DEG from the seq data would be invaluable. Can you show that there are increased anti-viral genes and link to the ISG data later? E.g. CD317 is not in your gene list on the volcano plot.

As noted above in response to Reviewer 1, point 6:

The scRNASeq data included in the manuscript was performed on samples isolated 3 days after SCV2 infection, and we therefore cannot assess how BCG influences ISG expression in the absence of SCV2 infection. To address this, we performed the flow cytometry experiments shown in Fig5, which provide evidence of surface protein expression on the epithelial cell types assessed.

Following the Reviewer's suggestion, we now include a new analysis of our scRNASeq data in FigS4B-C, where we use single cell gene set enrichment analysis (scGSEA) to show that anti-IFN γ treatment of iv BCG mice significantly reduces the enrichment of genes with validated anti-SCV2 activity in a number of different cell types despite viral loads being higher under these conditions. In contrast, no significant differences (except for B cells) were observed between PBS+SCV2+isotype versus PBS+SCV2+anti-IFN γ groups.

Further to this, we now also discuss a bioRxiv pre-print that became available after our submission where the Authors perform scRNASeq on lungs from control mice or animals inoculated iv with BCG 21 days prior (Lee et al, 2023, bioRxiv) (18). These data independently confirm our flow cytometry findings, showing that pulmonary epithelial cells express an anti-viral transcriptional signature after iv BCG. Together with our new analysis showing that IFN γ neutralization reduces the enrichment of anti-SCV2 genes, these data provide strong support for iv BCG induced IFN γ driving a broad anti-viral response in pulmonary epithelial cells.

The three ISGs (CD274, BST2 and Ly6A/E) were chosen for pursue further because they were among the differentially expressed genes and antibodies were commercially available to measure protein expression by flow cytometry.

Finally, CD317 (encoded by Bst2) is included in the CD45neg volcano plot in Fig3 as well as the DEG list in Table S8. To help avoid confusion, we have changed all reference to CD317 to BST2 throughout the manuscript.

Minor

1. Put the gene name (tetherin) for CD317 the first time it is mentioned (line 229) I think, that would make interpretation easier.

As above, CD317 is encoded by Bst2. We have changed all reference to CD317 to BST2 throughout the manuscript and have tetherin/CD317 in brackets following the first mention.

2. I think lines 311-315 are speculative. If you did the same model with influenza, I think you might see similar things.

The sentence has been re-phrased as follows: "It is this immunosuppressive property that potentially makes respiratory viruses, such as SCV2, amenable to restriction by pre-established IFN responses driven by concurrent or recent pulmonary infections."

3. Line 323 is not supported by the data – there is no data showing that IFN γ is replacing type 1 or 3 immunity. It is showing it can protect, but not that it replaces.

The sentence is now re-phased as follows: "Our findings reveal that, similar to type-1 and type-3 IFNs, pre-existing IFN γ can directly control SCV2 viral loads..."

4. In panels where it says No. DEG (e.g. 1E) it reads like no rather than number, can you rephrase

This has been changed to “DEGs between treatments”.

5. Figure 1I is very data dense and hard to interpret.

Figure 1I has been removed from the revised manuscript and replaced with flow cytometry data showing IFN γ protein expression by different cell types in response to comments by Reviewer 1. The panel showing Ifng transcript expression has been moved to FigS1G and is displayed as an UMAP overlay to aid with interpretation.

6. Figure 3B – not clear what significance is being compared to. Why doing fold change rather than actual values and show the control.

We chose to show the data as log₂ fold change over the PBS uninfected control as the raw values for each cytokine relative to each other differ greatly and the differences were not clear when shown as a heatmap. We now include all the raw data and statistical comparisons in FigS3.

The statistical comparison has been emphasized by adding “log₂FC over PBS, uninfected control” to the heading above the heatmap and included in the statistical analysis section for Fig3B.

7. Panel 5C – put Y axis labels on all graphs

This has been implemented.

References:

1. M. Rottman, C. Soudais, G. Vogt, L. Renia, J. F. Emile, H. Decaluwe, J. L. Gaillard, J. L. Casanova, IFN-gamma mediates the rejection of haematopoietic stem cells in IFN-gammaR1-deficient hosts. *PLoS Med* **5**, e26 (2008).
2. K. L. Hilligan, S. Namasivayam, C. S. Clancy, D. O'Mard, S. D. Oland, S. J. Robertson, P. J. Baker, E. Castro, N. L. Garza, B. A. P. Lafont, R. Johnson, F. Ronchese, K. D. Mayer-Barber, S. M. Best, A. Sher, Intravenous administration of BCG protects mice against lethal SARS-CoV-2 challenge. *J Exp Med* **219**, (2022).
3. C. H. Ladel, J. Hess, S. Daugelat, P. Mombaerts, S. Tonegawa, S. H. Kaufmann, Contribution of alpha/beta and gamma/delta T lymphocytes to immunity against Mycobacterium bovis bacillus Calmette Guerin: studies with T cell receptor-deficient mutant mice. *Eur J Immunol* **25**, 838-846 (1995).
4. T. S. Blackwell, J. W. Christman, The role of nuclear factor-kappa B in cytokine gene regulation. *Am J Respir Cell Mol Biol* **17**, 3-9 (1997).
5. N. L. Messina, S. Germano, R. McElroy, R. Rudraraju, R. Bonnici, L. F. Pittet, M. R. Neeland, S. Nicholson, K. Subbarao, N. Curtis, B. trial, Off-target effects of bacillus Calmette-Guerin vaccination on immune responses to SARS-CoV-2: implications for protection against severe COVID-19. *Clin Transl Immunology* **11**, e1387 (2022).
6. C. Counoupas, M. D. Johansen, A. O. Stella, D. H. Nguyen, A. L. Ferguson, A. Aggarwal, N. D. Bhattacharyya, A. Grey, O. Hutchings, K. Patel, R. Siddiquee, E. L. Stewart, C. G. Feng, N. G. Hansbro, U. Palendira, M. C. Steain, B. M. Saunders, J. K. K. Low, J. P. Mackay, A. D. Kelleher, W. J. Britton, S. G. Turville, P. M. Hansbro, J. A. Triccas, A single dose, BCG-adjuvanted COVID-19 vaccine provides sterilising immunity against SARS-CoV-2 infection. *NPJ Vaccines* **6**, 143 (2021).

7. S. Moorlag, E. Taks, T. Ten Doesschate, T. W. van der Vaart, A. B. Janssen, L. Muller, P. Ostermann, H. Dijkstra, H. Lemmers, E. Simonetti, M. Mazur, H. Schaal, R. Ter Heine, F. L. van de Veerdonk, C. P. Bleeker-Rovers, R. van Crevel, J. Ten Oever, M. I. de Jonge, M. J. Bonten, C. H. van Werkhoven, M. G. Netea, Efficacy of BCG Vaccination Against Respiratory Tract Infections in Older Adults During the Coronavirus Disease 2019 Pandemic. *Clin Infect Dis* **75**, e938-e946 (2022).
8. A. M. Blosssey, S. Bruckner, M. May, G. P. Parzmair, H. Sharma, U. Shaligram, L. Grode, S. H. E. Kaufmann, M. G. Netea, C. Schindler, VPM1002 as Prophylaxis Against Severe Respiratory Tract Infections Including Coronavirus Disease 2019 in the Elderly: A Phase 3 Randomized, Double-Blind, Placebo-Controlled, Multicenter Clinical Study. *Clin Infect Dis* **76**, 1304-1310 (2023).
9. T. Ten Doesschate, T. W. van der Vaart, P. A. Debisarun, E. Taks, S. Moorlag, N. Paternotte, W. G. Boersma, V. P. Kuiper, A. H. E. Roukens, B. J. A. Rijnders, A. Voss, K. M. Veerman, A. P. M. Kerckhoffs, J. T. Oever, R. van Crevel, C. van Nieuwkoop, A. Lalmohamed, J. van de Wijgert, M. G. Netea, M. J. M. Bonten, C. H. van Werkhoven, Bacillus Calmette-Guerin vaccine to reduce healthcare worker absenteeism in COVID-19 pandemic, a randomized controlled trial. *Clin Microbiol Infect* **28**, 1278-1285 (2022).
10. J. Claus, T. Ten Doesschate, C. Gumbs, C. H. van Werkhoven, T. W. van der Vaart, A. B. Janssen, G. Smits, R. van Binnendijk, F. van der Klis, D. van Baarle, F. L. Paganelli, H. Leavis, L. M. Verhagen, S. A. Joosten, M. J. M. Bonten, M. G. Netea, J. van de Wijgert, B. C.-C. S. Group, BCG Vaccination of Health Care Workers Does Not Reduce SARS-CoV-2 Infections nor Infection Severity or Duration: a Randomized Placebo-Controlled Trial. *mBio* **14**, e0035623 (2023).
11. L. F. Pittet, N. L. Messina, F. Orsini, C. L. Moore, V. Abruzzo, S. Barry, R. Bonnici, M. Bonten, J. Campbell, J. Croda, M. Dalcolmo, K. Gardiner, G. Gell, S. Germano, A. Gomes-Silva, C. Goodall, A. Gwee, T. Jamieson, B. Jardim, T. R. Kollmann, M. V. G. Lacerda, K. J. Lee, M. Lucas, D. J. Lynn, L. Manning, H. S. Marshall, E. McDonald, C. F. Munns, S. Nicholson, A. O'Connell, R. D. de Oliveira, S. Perlen, K. P. Perrett, C. Prat-Aymerich, P. C. Richmond, J. Rodriguez-Bano, G. Dos Santos, P. V. da Silva, J. W. Teo, P. Villanueva, A. Warris, N. J. Wood, A. Davidson, N. Curtis, B. T. C. Group, Randomized Trial of BCG Vaccine to Protect against Covid-19 in Health Care Workers. *N Engl J Med* **388**, 1582-1596 (2023).
12. E. L. Koekenbier, K. Fohse, J. S. van de Maat, J. J. Oosterheert, C. van Nieuwkoop, J. J. Hoogerwerf, M. P. Grobusch, M. van den Bosch, J. H. H. van de Wijgert, M. G. Netea, F. R. Rosendaal, M. J. M. Bonten, C. Werkhoven, B.-P. s. group, Bacillus Calmette-Guerin vaccine for prevention of COVID-19 and other respiratory tract infections in older adults with comorbidities: a randomized controlled trial. *Clin Microbiol Infect*, (2023).
13. L. R. B. Dos Anjos, A. C. da Costa, A. Cardoso, R. A. Guimaraes, R. L. Rodrigues, K. M. Ribeiro, K. C. M. Borges, A. C. O. Carvalho, C. I. S. Dias, A. O. Rezende, C. C. Souza, R. R. M. Ferreira, G. Saraiva, L. C. S. Barbosa, T. D. S. Vieira, M. B. Conte, M. F. Rabahi, A. Kipnis, A. P. Junqueira-Kipnis, Efficacy and Safety of BCG Revaccination With *M. bovis* BCG Moscow to Prevent COVID-19 Infection in Health Care Workers: A Randomized Phase II Clinical Trial. *Front Immunol* **13**, 841868 (2022).
14. A. P. Santos, G. L. Werneck, A. P. R. Dalvi, C. C. Dos Santos, P. Tierno, H. S. Condello, B. Macedo, J. A. de Medeiros Leung, J. de Souza Nogueira, L. Malvao, R. Galliez, R. Aguiar, R. Stefan, S. M. Knackfuss, E. C. da Silva, T. Castineiras, R. de Andrade Medronho, E. S. JRL, R. L. R. Alves, L. C. de Moraes Sobrino Porto, L. S. Rodrigues, A. L. Kritski, F. C. de Queiroz Mello, The effect of BCG vaccination on infection and antibody levels against SARS-CoV-2-The results of ProBCG: a multicenter randomized clinical trial in Brazil. *Int J Infect Dis* **130**, 8-16 (2023).
15. D. A. Kaveh, M. C. Garcia-Pelayo, P. J. Hogarth, Persistent BCG bacilli perpetuate CD4 T effector memory and optimal protection against tuberculosis. *Vaccine* **32**, 6911-6918 (2014).
16. A. M. Minassian, I. Satti, I. D. Poulton, J. Meyer, A. V. Hill, H. McShane, A human challenge model for *Mycobacterium tuberculosis* using *Mycobacterium bovis* bacille Calmette-Guerin. *J Infect Dis* **205**, 1035-1042 (2012).
17. B. Freyne, N. L. Messina, S. Donath, S. Germano, R. Bonnici, K. Gardiner, D. Casalaz, R. M. Robins-Browne, M. G. Netea, K. L. Flanagan, T. Kollmann, N. Curtis, B. C. G. f. A. Melbourne Infant Study, G. Infection Reduction, Neonatal BCG Vaccination Reduces Interferon-gamma Responsiveness to Heterologous Pathogens in Infants From a Randomized Controlled Trial. *J Infect Dis* **221**, 1999-2009 (2020).

18. A. Lee, K. Floyd, S.-Y. Wu, Z. Fang, T. K. Tan, C. Li, H. Hui, D. Scoville, A. Ruggiero, Y. Liang, A. Pavenko, V. Lujan, G. P. Nolan, P. Arunachalam, M. Suthar, B. Pulendran, Integrated Organ Immunity: Antigen-specific CD4-T cell-derived IFN- γ induced by BCG imprints prolonged lung innate resistance against respiratory viruses. *bioRxiv*, 2023.2007.2031.551354 (2023).
19. K. L. Hilligan, S. Namasivayam, A. Sher, BCG mediated protection of the lung against experimental SARS-CoV-2 infection. *Front Immunol* **14**, 1232764 (2023).

Reviewers' Comments:

Reviewer #1:

Remarks to the Author:

My concerns have been adequately addressed in the revised version.

Reviewer #2:

Remarks to the Author:

thanks for making the corrections